# Structure-free Graph Condensation: From Large-scale Graphs to Condensed Graph-free Data

**Xin Zheng**[1], **Miao Zhang**[2], **Chunyang Chen**[1], **Quoc Viet Hung Nguyen**[3], **Xingquan Zhu**[4], **Shirui Pan**[3][†]

[1]Monash University, Australia,  [2]Harbin Institute of Technology (Shenzhen), China
[3]Griffith University, Australia,  [4]Florida Atlantic University, USA
xin.zheng@monash.edu, zhangmiao@hit.edu.cn, chunyang.chen@monash.edu
henry.nguyen@griffith.edu.au, xzhu3@fau.edu, s.pan@griffith.edu.au

## Abstract

Graph condensation, which reduces the size of a large-scale graph by synthesizing a small-scale condensed graph as its substitution, has immediate benefits for various graph learning tasks. However, existing graph condensation methods rely on the joint optimization of nodes and structures in the condensed graph, and overlook critical issues in effectiveness and generalization ability. In this paper, we advocate a new Structure-Free Graph Condensation paradigm, named SFGC, to distill a large-scale graph into a small-scale graph node set without explicit graph structures, *i.e.*, graph-free data. Our idea is to implicitly encode topology structure information into the node attributes in the synthesized graph-free data, whose topology is reduced to an identity matrix. Specifically, SFGC contains two collaborative components: (1) a training trajectory meta-matching scheme for effectively synthesizing small-scale graph-free data; (2) a graph neural feature score metric for dynamically evaluating the quality of the condensed data. Through training trajectory meta-matching, SFGC aligns the long-term GNN learning behaviors between the large-scale graph and the condensed small-scale graph-free data, ensuring comprehensive and compact transfer of informative knowledge to the graph-free data. Afterward, the underlying condensed graph-free data would be dynamically evaluated with the graph neural feature score, which is a closed-form metric for ensuring the excellent expressiveness of the condensed graph-free data. Extensive experiments verify the superiority of SFGC across different condensation ratios.[‡]

## 1 Introduction

As prevalent graph data learning models, graph neural networks (GNNs) have attracted much attention and achieved great success [68, 36, 22, 82, 25, 24, 23]. Various graph data in the real world comprises millions of nodes and edges[37, 38, 43], reflecting diverse node attributes and complex structural connections [33, 34, 32, 54]. Modeling such large-scale graphs brings serious challenges in both data storage and GNN model designs, hindering the applications of GNNs in many industrial scenarios [74, 2, 64, 73, 80, 71, 72]. For instance, designing GNN models usually requires repeatedly training GNNs for adjusting proper hyper-parameters and constructing optimal model architectures. When taking large-scale graphs as training data, repeated training through message passing along complex graph structures, makes it highly computation-intensive and time-consuming through try-and-error.

To address these challenges brought by the scale of graph data, a natural data-centric solution [78] is graph size reduction, which transforms the real-world large-scale graph to a small-scale

---

[†]Corresponding author
[‡]Code is available at https://github.com/Amanda-Zheng/SFGC

37th Conference on Neural Information Processing Systems (NeurIPS 2023).

graph, such as graph sampling [66, 6], graph coreset [47, 60], graph sparsification [1, 5], and graph coarsening [3, 28]. These conventional methods either extract representative nodes and edges or preserve specific graph properties from the large-scale graphs, resulting in severe limitations of the obtained small-scale graphs in the following two folds. First, the available information on derived small-scale graphs is significantly upper-bounded and limited within the range of large-scale graphs [66, 60]. Second, the preserved properties of small-scale graphs, *e.g.*, spectrum and clustering, might not always be optimal for training GNNs for downstream tasks [1, 3, 28].

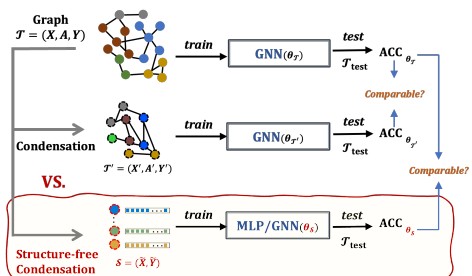

Figure 1: Comparisons of condensation *vs* structure-free condensation on graphs.

In light of these limitations of conventional methods, in this work, we mainly focus on graph condensation [27, 26], a new rising synthetic method for graph size reduction. Concretely, graph condensation aims to directly optimize and synthesize a small-scale condensed graph, so that the small-scale condensed graph could achieve comparable test performance as the large-scale graph when training the same GNN model. Therefore, the principal goal of graph condensation is to ensure consistent test results for GNNs when taking the large-scale graph and the small-scale condensed graph as training data.

However, due to the structural characteristic of graph data, nodes and edges are tightly coupled. This makes condensing graph data a complicated task since high-quality condensed graphs are required to jointly synthesize discriminative node attributes and topology structures. Some recent works have made initial explorations of graph condensation [27, 26]. For instance, GCOND [27] proposed the online gradient matching schema between the synthesized small-scale graph and the large-scale graph, followed by a condensed graph structure learning module for synthesizing both condensed nodes and structures. However, existing methods overlook two-fold critical issues regarding **effectiveness and generalization ability**. First, graph condensation requires a triple-level optimization to jointly learn three objectives: GNN parameters, distilled node attributes, and topology structures. Such complex optimization cannot guarantee optimal solutions for both nodes and edges in the condensed graph, significantly limiting its effectiveness as the representative of the large-scale graph. Furthermore, existing online GNN gradients [27, 26] are calculated with the short-range matching, leading to the short-sight issue of failing to imitate holistic GNN learning behaviors, limiting the quality of condensed graphs. Second, existing condensed graphs generally show poor generalization ability across different GNN models [27, 26], because different GNN models vary in their convolution operations along graph structures. As a result, existing methods are vulnerable to overfiting of specific GNN architectures by distilling convolutional information into condensed graph structures.

To deal with the above two-fold challenges, in this work, we propose a novel Structure-Free Graph Condensation paradigm, named SFGC, to distill large-scale real-world graphs into small-scale synthetic graph node sets without graph structures, *i.e.*, condensed graph-free data. Different from conventional graph condensation that synthesizes both nodes and structures to derive a small-scale graph, as shown in Fig. 1, the proposed structure-free graph condensation only synthesizes a small-scaled node set to train a GNN/MLP, when it implicitly encodes topology structure information into the node attributes in the synthesized graph-free data, by simplifying the condensed topology to an identity matrix. Overall, the proposed SFGC contains two essential components: (1) a training trajectory meta-matching scheme for effectively synthesizing small-scale graph-free data; (2) a graph neural feature score metric for dynamically evaluating the quality of condensed graph-free data. To address the short-sight issue of existing online gradient matching, our training trajectory meta-matching scheme first trains a set of training trajectories of GNNs on the large-scale graph to acquire an expert parameter distribution, which serves as offline guidance for optimizing the condensed graph-free data. Then, the proposed SFGC conducts meta-matching to align the long-term GNN learning behaviors between the large-scale graph and condensed graph-free data by sampling from the training trajectory distribution, enabling the comprehensive and compact transfer of informative knowledge to the graph-free data. At each meta-matching step, we would obtain updated condensed graph-free data, which would be fed into the proposed graph neural feature score metric for dynamically evaluating its quality. This metric is derived based on the closed-form solutions of GNNs under the graph neural tangent kernel (GNTK) ridge regression, eliminating the iterative training

of GNNs in the dynamic evaluation. Finally, the proposed `SFGC` selects the condensed graph-free data with the smallest score as the optimal representative of the large-scale graph. Our proposed structure-free graph condensation method could benefit many potential application scenarios, such as, *graph neural architecture search* [79, 81], *privacy protection* [69], *adversarial robustness* [67, 70], *continual learning* [78], and so on. We provide detailed demonstrations of how our method facilitates the development of these areas in Appendix B

In summary, the contributions of this work are listed as follows:

- We propose a novel Structure-Free Graph Condensation paradigm to effectively distill large-scale real-world graphs to small-scale synthetic graph-free data with superior expressiveness, to the best of our knowledge, for the first time.

- To explicitly imitate the holistic GNN training process, we propose the training trajectory meta-matching scheme, which aligns the long-term GNN learning behaviors between the large-scale graph and the condensed graph-free data, with the theoretical guarantee of eliminating graph structure constraints.

- To ensure the high quality of the condensed data, we derive a GNTK-based graph neural feature score metric, which dynamically evaluates the small-scale graph-free data at each meta-matching step and selects the optimal one. Extensive experiments verify the superiority of our method.

**Prior Works.** Our research falls into the research topic *dataset distillation (condensation)* [30, 59], which aims to synthesize a small typical dataset that distills the most important knowledge from a given large target dataset as its effective substitution. Considering most of the works condense image data [59, 39, 77, 76, 4], due to the complexity of graph structural data, only a few works [27, 26] address graph condensation, while our research designs a new structure-free graph condensation paradigm for addressing the effectiveness and generalization ability issues in existing graph condensation works. Our research also significantly differs from other general graph size reduction methods, for instance, graph coreset [47, 60], graph sparsification [1, 5] and so on. More detailed discussions about related works can be found in Appendix A.

## 2 Structure-Free Graph Condensation

### 2.1 Preliminaries

**Notations.** Denote a large-scale graph dataset to be condensed by $\mathcal{T} = (\mathbf{X}, \mathbf{A}, \mathbf{Y})$, where $\mathbf{X} \in \mathbb{R}^{N \times d}$ denotes $N$ number of nodes with $d$-dimensional features, $\mathbf{A} \in \mathbb{R}^{N \times N}$ denotes the adjacency matrix indicating the edge connections, and $\mathbf{Y} \in \mathbb{R}^{N \times C}$ denotes the $C$-classes of node labels. In general, graph condensation synthesizes a small-scale graph dataset denoted as $\mathcal{T}' = (\mathbf{X}', \mathbf{A}', \mathbf{Y}')$ with $\mathbf{X}' \in \mathbb{R}^{N' \times d}$, $\mathbf{A}' \in \mathbb{R}^{N' \times N'}$, and $\mathbf{Y}' \in \mathbb{R}^{N' \times C}$ when $N' \ll N$. In this work, we propose the structure-free graph condensation paradigm, which aims to synthesize a small-scale graph node set $\mathcal{S} = (\widetilde{\mathbf{X}}, \widetilde{\mathbf{Y}})$ without explicitly condensing graph structures, *i.e.*, the condensed graph-free data, as an effective substitution of the given large-scale graph. Hence, $\widetilde{\mathbf{X}}$ contains joint node context attributes and topology structure information, which is a more compact representative compared with $(\mathbf{X}', \mathbf{A}')$.

**Graph Condensation.** Given a GNN model parameterized by $\boldsymbol{\theta}$, graph condensation [27] is defined to solve the following triple-level optimization objective by taking $\mathcal{T} = (\mathbf{X}, \mathbf{A}, \mathbf{Y})$ as input:

$$\min_{\mathcal{T}'} \mathcal{L} \left[ \mathrm{GNN}_{\boldsymbol{\theta}_{\mathcal{T}'}}(\mathbf{X}, \mathbf{A}), \mathbf{Y} \right]$$

$$s.t. \quad \boldsymbol{\theta}_{\mathcal{T}'} = \arg \min_{\boldsymbol{\theta}} \mathcal{L} \left[ \mathrm{GNN}_{\boldsymbol{\theta}} \left( \mathbf{X}', \mathbf{A}' \right), \mathbf{Y}' \right], \tag{1}$$

$$\boldsymbol{\psi}_{\mathbf{A}'} = \arg \min_{\boldsymbol{\psi}} \mathcal{L} \left[ \mathrm{GSL}_{\boldsymbol{\psi}} \left( \mathbf{X}' \right) \right],$$

where $\mathrm{GSL}_{\boldsymbol{\psi}}$ is a submodule parameterized by $\boldsymbol{\psi}$ to synthesize the graph structure $\mathbf{A}'$. One of inner loops learns the optimal GNN parameters $\boldsymbol{\theta}_{\mathcal{T}'}$, while another learns the optimal GSL parameters $\boldsymbol{\psi}_{\mathbf{A}'}$ to obtain the condensed $\mathbf{A}'$, and the outer loop updates the condensed nodes $\mathbf{X}'$. All these comprise the condensed small-scale graph $\mathcal{T}' = (\mathbf{X}', \mathbf{A}', \mathbf{Y}')$, where $\mathbf{Y}'$ is pre-defined based on the class distribution of the label space $\mathbf{Y}$ in the large-scale graph.

Overall, the above optimization objective needs to solve the following variables iteratively: (1) condensed $\mathbf{X}'$; (2) condensed $\mathbf{A}'$ with $\mathrm{GSL}_{\boldsymbol{\psi}_{\mathbf{A}'}}$; and (3) $\mathrm{GNN}_{\boldsymbol{\theta}_{\mathcal{T}'}}$. Jointly learning these interdependent

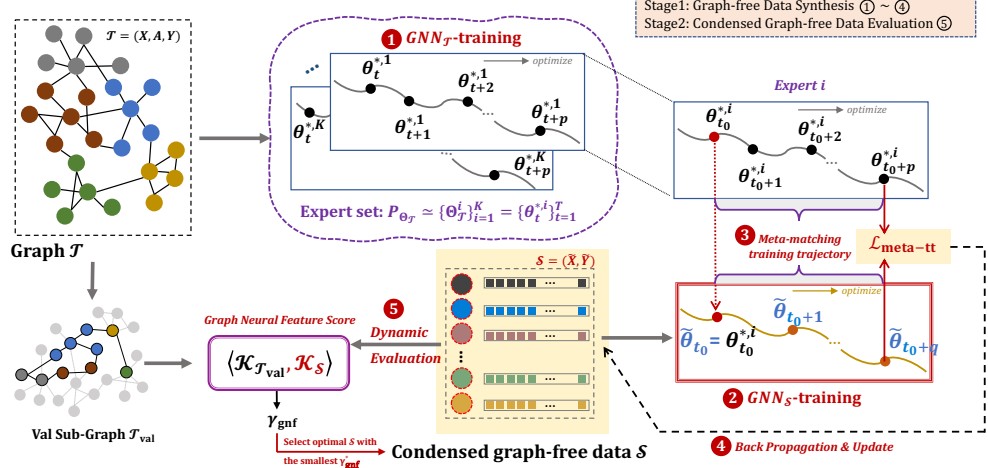

Figure 2: Overall pipeline of the proposed Structure-Free Graph Condensation (SFGC) framework.

objectives is highly challenging. It is hard to guarantee that each objective obtains the optimal and convergent solution in such a complex and nested optimization process, resulting in the limited expressiveness of the condensed graph. This dilemma motivates us to reconsider the optimization objective of graph condensation to synthesize the condensed graph more effectively.

**Graph Neural Tangent Kernel (GNTK).** As a new class of graph kernels, graph neural tangent kernel (GNTK) is easy to train with provable theoretical guarantees, and meanwhile, enjoys the full expressive power of GNNs [10, 19, 21, 41]. In general, GNTK can be taken as the infinitely-wide multi-layer GNNs trained by gradient descent. It learns a class of smooth functions on graphs with close-form solutions. More specifically, let $G = (V, E)$ denote a graph with nodes $V$ and edges $E$, where each node $v \in V$ within its neighbor set $\mathcal{N}(v)$. Given two graphs $G = (V, E)$ and $G' = (V', E')$ with $n$ and $n'$ number of nodes, their covariance matrix between input features can be denoted as $\Sigma^{(0)}(G, G') \in \mathbb{R}^{n \times n'}$. Each element in $\left[\Sigma^{(0)}(G, G')\right]_{uu'}$ is the inner product $\boldsymbol{h}_u^\top \boldsymbol{h}_{u'}$, where $\boldsymbol{h}_u$ and $\boldsymbol{h}_{u'}$ are of input features of two nodes $u \in V$ and $u' \in V'$. Then, for each GNN layer $\ell \in \{0, 1, \ldots, L\}$ that has $\mathcal{B}$ fully-connected layers with ReLU activation, GNTK calculates $\mathcal{K}_{(\beta)}^{(\ell)}\langle G, G'\rangle$ for each $\beta \in [\mathcal{B}]$:

$$
\begin{aligned}
\left[\mathcal{K}_{(\beta)}^{(\ell)}\langle G, G'\rangle\right]_{uu'} &= \left[\mathcal{K}_{(\beta-1)}^{(\ell)}\langle G, G'\rangle\right]_{uu'}\left[\dot{\boldsymbol{\Sigma}}_{(\beta)}^{(\ell)}(G, G')\right]_{uu'} \\
&\quad + \left[\boldsymbol{\Sigma}_{(\beta)}^{(\ell)}(G, G')\right]_{uu'},
\end{aligned}
\tag{2}
$$

where $\dot{\Sigma}^{(\ell)}$ denotes the derivative *w.r.t.* the $\ell$-th GNN layer of the covariance matrix, and the $(\ell+1)$-th layer's covariance matrix aggregates neighbors along graph structures as $\left[\boldsymbol{\Sigma}_{(0)}^{(\ell+1)}(G, G')\right]_{uu'} = \sum_{v \in \mathcal{N}(u) \cup \{u\}} \sum_{v' \in \mathcal{N}(u') \cup \{u'\}} \left[\boldsymbol{\Sigma}_{(\mathcal{B})}^{(\ell)}(G, G')\right]_{vv'}$, ditto for the kernel $\left[\mathcal{K}_{(0)}^{(\ell+1)}(G, G')\right]_{uu'}$. With the GNTK matrix $\mathcal{K}_{(\mathcal{B})}^{(L)}\langle G, G'\rangle \in \mathbb{R}^{n \times n'}$ at the node level, we use the graph kernel method to solve the equivalent GNN model for node classification with closed-form solutions. This would significantly benefit the efficiency of condensed data evaluation by eliminating iterative GNN training.

## 2.2 Overview of SFGC Framework

The crux of achieving structure-free graph condensation is in determining discriminative node attribute contexts, which implicitly integrates topology structure information. We compare the paradigms between existing graph condensation (GC) and our new structure-free condensation SFGC as follows:

$$
\begin{aligned}
\mathcal{T} = (\mathbf{X}, \mathbf{A}, \mathbf{Y}) &\to \mathcal{T}' = (\mathbf{X}', \mathbf{A}', \mathbf{Y}'), \quad \text{GC.} \\
\mathcal{T} = (\mathbf{X}, \mathbf{A}, \mathbf{Y}) &\to \mathcal{S} = (\widetilde{\mathbf{X}}, \mathbf{I}, \widetilde{\mathbf{Y}}) = \mathcal{S} = (\widetilde{\mathbf{X}}, \widetilde{\mathbf{Y}}), \quad \text{SFGC.}
\end{aligned}
\tag{3}
$$

It is fundamentally different between existing GC paradigm with $\mathcal{T} \rightarrow \mathcal{T}'$ and our SFGC with $\mathcal{T} \rightarrow \mathcal{S}$. Our idea is to synthesize a graph-free data $\mathcal{S} = (\widetilde{\mathbf{X}}, \mathbf{I}, \widetilde{\mathbf{Y}})$ whose topology is reduced to an identity matrix $\mathbf{I}$ (*i.e.*, structure-free), instead of explicitly learning $\mathbf{A}'$ as existing GC. To enforce node attribute $\widetilde{\mathbf{X}}$ encoding topology structure information as $\widetilde{\mathbf{X}} \simeq (\mathbf{X}, \mathbf{A})$, we propose a long-term imitation learning process with training trajectories, which requires a GNN model learned from $\mathcal{S}$, *i.e.*, $\text{GNN}_{\mathcal{S}} = \text{GNN}(\widetilde{\mathbf{X}}, \mathbf{I}, \widetilde{\mathbf{Y}})$ must imitate a GNN model from the original graph, *i.e.*, $\text{GNN}_{\mathcal{T}} = \text{GNN}(\mathbf{X}, \mathbf{A}, \mathbf{Y})$. We provide more theoretical illustrations of the proposed structure-free graph condensation paradigm from the views of statistical learning and information flow, respectively, in Appendix D. The overall pipeline of the proposed Structure-Free Graph Condensation (SFGC) framework is shown in Fig. 2.

We consider two coupled stages in the proposed SFGC framework, *i.e.*, graph-free data synthesis ( ① $\sim$ ④ ) and condensed graph-free data evaluation ( ⑤ ), corresponding to two essential components: (1) the training trajectory meta-matching scheme and (2) the graph neural feature score metric. Concretely, as illustrated in Fig. 2, taking the given large-scale graph $\mathcal{T}$ as input, we first ( ① ) train an expert set of $\text{GNN}_{\mathcal{T}}$, parameterized by $\left\{ \mathbf{\Theta}_{\mathcal{T}}^{i} \right\}_{i=1}^{K} = \left\{ \boldsymbol{\theta}_{t}^{*,i} \right\}_{t=1}^{T}$, denoting $K$ numbers of expert training trajectories, each within $T$ time steps. Then, we sample a single expert training trajectory $i$ at $t_0$ step, *i.e.*, $\boldsymbol{\theta}_{t_0}^{*,i}$, and further use it to initialize the model ( ② ) $\text{GNN}_{\mathcal{S}}$ trained by the condensed graph-free data $\mathcal{S}$. Then, after $q$ steps of $\text{GNN}_{\mathcal{S}}$ and $p$ steps of $\text{GNN}_{\mathcal{T}}$, we conduct the meta-matching ( ③ ) between the long-term intervals of training trajectories $\left[ \widetilde{\boldsymbol{\theta}}_{t_0}, \widetilde{\boldsymbol{\theta}}_{t_0+q} \right]$ and $\left[ \boldsymbol{\theta}_{t_0}^{*,i}, \boldsymbol{\theta}_{t_0+p}^{*,i} \right]$ with the proposed meta-matching loss $\mathcal{L}_{\text{meta-tt}}$. Next, the loss function back-propagates along $\text{GNN}_{\mathcal{S}}$ and the condensed graph-free data $\mathcal{S}$ is updated ( ④ ). After, the updated $\mathcal{S}$ at the current step is used to calculate the GNTK-based graph neural feature score metric $\gamma_{\text{gnf}}$ for dynamic evaluation ( ⑤ ), along with the large-scale validation subgraph $\mathcal{T}_{\text{val}}$. Finally, SFGC selects the optimal condensed graph-free data with the smallest $\gamma_{\text{gnf}}^{*}$ as the expressive substitution of the large-scale graph.

## 2.3 Training Trajectory Meta-matching

Different from existing graph condensation methods [27, 26] that conduct online gradient matching within the short range, *i.e.*, step-by-step or single-step matching gradients between the large-scale graph and the condensed graph, the proposed SFGC matches their long-term GNN training trajectories with the guidance of the offline expert parameter distribution. Concretely, inspired by [4], we first train $K$ numbers of GNN models with the same architecture denoted as $\text{GNN}_{\mathcal{T}}$ on the large-scale graph $\mathcal{T}$. Then, we save their network parameters $\left\{ \mathbf{\Theta}_{\mathcal{T}}^{i} \right\}_{i=1}^{K} = \left\{ \boldsymbol{\theta}_{t}^{*,i} \right\}_{t=1}^{T}$ at certain epoch intervals, resulting in $K$ numbers of expert training trajectories that have comprehensive knowledge of the large-scale graph in terms of GNN's training process. Such expert training trajectories further build a parameter distribution $P_{\mathbf{\Theta}_{\mathcal{T}}}$ denoting the GNN learning behavior on the large-scale graph. Note that such a parameter distribution is pre-calculated and stored. Hence, this process can be offline and separated from the end-to-end graph condensation pipeline, moderately reducing the online computation costs.

By sampling from the pre-derived parameter distribution $P_{\mathbf{\Theta}_{\mathcal{T}}}$, we optimize the following objective for synthesizing $\mathcal{S}$ as:

$$\min_{\mathcal{S}} \mathrm{E}_{\boldsymbol{\theta}_{t}^{*,i} \sim P_{\mathbf{\Theta}_{\mathcal{T}}}} \left[ \mathcal{L}_{\text{meta-tt}} \left( \boldsymbol{\theta}_{t}^{*} |_{t=t_0}^{p}, \widetilde{\boldsymbol{\theta}}_{t} |_{t=t_0}^{q} \right) \right]. \tag{4}$$

Note that $\boldsymbol{\theta}_{t}^{*} |_{t=t_0}^{p}$ and $\widetilde{\boldsymbol{\theta}}_{t} |_{t=t_0}^{q}$ denote the parameters of $\text{GNN}_{\mathcal{T}}$ within the range of $(t_0, t_0 + p)$ and $\text{GNN}_{\mathcal{S}}$ within $(t_0, t_0 + q)$, respectively, where $t_0 < t_0 + p < T$. More specifically, $\mathcal{L}_{\text{meta-tt}}$ calculates certain parameter training intervals within $\left[ \boldsymbol{\theta}_{t_0}^{*,i}, \boldsymbol{\theta}_{t_0+p}^{*,i} \right]$ and $\left[ \widetilde{\boldsymbol{\theta}}_{t_0}, \widetilde{\boldsymbol{\theta}}_{t_0+q} \right]$ as

$$\mathcal{L}_{\text{meta-tt}} = \frac{\left\| \widetilde{\boldsymbol{\theta}}_{t_0+q} - \boldsymbol{\theta}_{t_0+p}^{*,i} \right\|_{2}^{2}}{\left\| \widetilde{\boldsymbol{\theta}}_{t_0} - \boldsymbol{\theta}_{t_0+p}^{*,i} \right\|_{2}^{2}}. \tag{5}$$

Here, we initialize the parameter of $\text{GNN}_{\mathcal{S}}$ with that of $\text{GNN}_{\mathcal{T}}$ at $t_0$ training step, so that we have $\boldsymbol{\theta}_{t_0}^{*,i} = \widetilde{\boldsymbol{\theta}}_{t_0}$. And we consider the expectation on $\mathcal{S}$ *w.r.t.* different initialized parameter $\boldsymbol{\theta}_{t_0}^{*,i}$ in

**Algorithm 1** Structure-Free Graph Condensation (SFGC)

**Require:** (1) $P_{\mathbf{\Theta}_{\mathcal{T}}}$: Pretrained a set of $K$ numbers of GNNs on the large-scale graph $\mathrm{GNN}_{\mathcal{T}}$ parameterized by $\mathbf{\Theta}_{\mathcal{T}}$; (2) $T_0$: numbers of meta-matching steps; (3) $T_1$: $\mathrm{GNN}_{\mathcal{S}}$ training steps.

**Ensure:** A small-scale condensed graph-free data $\mathcal{S} = \left( \widetilde{\mathbf{X}}, \widetilde{\mathbf{Y}} \right)$

1: **while** $\eta < T_0$ **do**
2:   randomly sample a pretrained training trajectory in $P_{\mathbf{\Theta}_{\mathcal{T}}}$ and calculate the $\mathcal{L}_{\text{meta-tt}}$ according to Eq. (5);
3:   **for** $t < T_1$ **do**
4:     train $\mathrm{GNN}_{\mathcal{S}}$ through gradient descent by $\widetilde{\boldsymbol{\theta}}_{\mathcal{S}}^{t+1} \leftarrow \widetilde{\boldsymbol{\theta}}_{\mathcal{S}}^t - \zeta \nabla_{\widetilde{\boldsymbol{\theta}}} \mathcal{L}_{\text{cls}} \left[ \mathrm{GNN}_{\mathcal{S}} \left( \widetilde{\mathbf{X}}, \mathbf{I} \right), \widetilde{\mathbf{Y}} \right]$, where $\zeta$ is the step size;
5:   **end for**
6:   update the condensed graph-free data $\mathcal{S}_{\eta}$ according to Eq. (4);
7:   calculate current $\eta$-th step $\gamma_{\text{gnf}}(\mathcal{S}_{\eta})$ according to Eq. (7);
8: **end while**
9: select the optimal condensed graph-free data $\mathcal{S}_{\eta}^*$ with the smallest score $\gamma_{\text{gnf}}^*$ as the final output.

distribution $P_{\mathbf{\Theta}_{\mathcal{T}}}$, which can be taken as a 'meta' way to make the distilled dataset $\mathcal{S}$ adapt different parameter initialization. That is why we call it 'meta-matching'. In this way, when initializing $\mathrm{GNN}_{\mathcal{T}}$ and $\mathrm{GNN}_{\mathcal{S}}$ with the same model parameters, Eq. (5) contributes to aligning the learning behaviors of $\mathrm{GNN}_{\mathcal{T}}$ that experiences $p$-steps optimization, to $\mathrm{GNN}_{\mathcal{S}}$ that experiences $q$-steps optimization. In this way, the proposed training trajectory meta-matching schema could comprehensively imitate the long-term learning behavior of GNN training. As a result, the informative knowledge of the large-scale graph $\mathcal{T}$ can be effectively transferred to the small-scale condensed graph-free data $\mathcal{S} = (\widetilde{\mathbf{X}}, \widetilde{\mathbf{Y}})$ in the above outer-loop optimization objective of Eq. (4).

For the inner loop, we train $\mathrm{GNN}_{\mathcal{S}}$ on the synthesized small-scale condensed graph-free data for optimizing its model parameter until the optimal $\widetilde{\boldsymbol{\theta}}_{\mathcal{S}}^*$. Therefore, the final optimization objective of the proposed SFGC is

$$\min_{\mathcal{S}} \mathrm{E}_{\boldsymbol{\theta}_t^{*,i} \sim P_{\mathbf{\Theta}_{\mathcal{T}}}} \left[ \mathcal{L}_{\text{meta-tt}} \left( \boldsymbol{\theta}_t^* |_{t=t_0}^p, \widetilde{\boldsymbol{\theta}}_t |_{t=t_0}^q \right) \right],$$
$$s.t. \quad \widetilde{\boldsymbol{\theta}}_{\mathcal{S}}^* = \arg\min_{\widetilde{\boldsymbol{\theta}}} \mathcal{L}_{\text{cls}} \left[ \mathrm{GNN}_{\widetilde{\boldsymbol{\theta}}} \left( \mathcal{S} \right) \right],$$

(6)

where $\mathcal{L}_{\text{cls}}$ is the node classification loss calculated with the cross-entropy on graphs. Compared with the triple-level optimization in Eq. (1), the proposed SFGC directly replaces the learnable $\mathbf{A}'$ in Eq. (1) with a fixed identity matrix $\mathbf{I}$, resulting in the condensed structure-free graph data $\mathcal{S} = (\widetilde{\mathbf{X}}, \mathbf{I}, \widetilde{\mathbf{Y}})$. Without synthesizing condensed graph structures with $\mathrm{GSL}_{\psi}$, the proposed SFGC refines the complex triple-level optimization to the bi-level one, ensuring effectiveness of the condensed graph-free data.

Hence, the advances of the training trajectory meta-matching schema in the proposed SFGC can be summarized as follows: (1) compared with the online gradient calculation, SFGC's offline parameter sampling avoids dynamically computing and storing gradients of both the large and condensed small graphs, reducing computation and memory costs during the condensation process; (2) compared with short-range matching, SFGC's long-term meta-matching avoids condensed data to short-sightedly fit certain optimization steps, contributing to a more holistic and comprehensive way to imitate GNN's learning behaviors.

## 2.4 Graph Neural Feature Score

For each update of the outer loop in Eq. (6), we would synthesize the brand-new condensed graph-free data. However, evaluating the quality of the underlying condensed graph-free data in the dynamical meta-matching condensation process is quite challenging. That is because we cannot quantity a graph dataset's performance without blending it in a GNN model. And the condensed graph-free data itself cannot be measured by convergence or decision boundary. Generally, to evaluate the condensed graph-free data, we use it to train a GNN model. If the condensed data at a certain meta-matching step achieves better GNN test performance on node classification, it indicates the higher quality of

the current condensed data. That means, evaluating condensed graph-free data needs an extra process of training a GNN model from scratch, leading to much more time and computation costs.

In light of this, we aim to derive a metric to dynamically evaluate the condensed graph-free data in the meta-matching process, and utilize it to select the optimal small-scale graph-free data. Meanwhile, the evaluation progress would not introduce extra GNN iterative training for saving computation and time. To achieve this goal, we first identify what characteristics such a metric should have: (1) closed-form solutions of GNNs to avoid iterative training in evaluation; (2) the ability to build strong connections between the large-scale graph and the small-scale synthesized graph-free data. In this case, the graph neural tangent kernel (GNTK) stands out, as a typical class of graph kernels, and has the full expressive power of GNNs with provable closed-form solutions. Moreover, as shown in Eq. (2), GNTK naturally builds connections between arbitrary two graphs even with different sizes.

Based on the graph kernel method with GNTK, we proposed a graph neural feature score metric $\gamma_{\text{gnf}}$ to dynamically evaluate and select the optimal condensed graph-free data as follows:

$$\gamma_{\text{gnf}}(\mathcal{S}) = \frac{1}{2} \left\| \mathbf{Y}_{\text{val}} - \mathcal{K} \left\langle \mathcal{T}_{\text{val}}, \mathcal{S} \right\rangle \left( \mathcal{K} \left\langle \mathcal{S}, \mathcal{S} \right\rangle + \lambda \mathbf{I} \right)^{-1} \widetilde{\mathbf{Y}} \right\|^2, \tag{7}$$

where $\mathcal{K} \left\langle \mathcal{T}_{\text{val}}, \mathcal{S} \right\rangle \in \mathbb{R}^{N_{\text{val}} \times N'}$ and $\mathcal{K} \left\langle \mathcal{S}, \mathcal{S} \right\rangle \in \mathbb{R}^{N' \times N'}$ denote the node-level GNTK matrices derived according to Eq. (2). And $\mathcal{T}_{\text{val}}$ is the validation sub-graph of the large-scale graph with $N_{\text{val}}$ numbers of nodes. Concretely, $\gamma_{\text{gnf}}(\mathcal{S})$ calculates the graph neural tangent kernel based ridge regression error. It measures that, given an infinitely-wide GNN trained on the condensed graph $\mathcal{S}$ with ridge regression, how close such GNN's prediction on $\mathcal{T}_{\text{val}}$ to its ground truth labels $\mathbf{Y}_{\text{val}}$. Note that Eq. (7) can be regarded as the Kernel Inducing Point (KIP) algorithm [39, 40] adapted to the GNTK kernel on GNN models.

Hence, the proposed graph neural feature score meets the above-mentioned characteristics as: (1) it calculates a closed-form GNTK-based ridge regression error for evaluation without iteratively training GNN models; (2) it strongly connects the condensed graph-free data with the large-scale validation graph; In summary, the overall algorithm of the proposed SFGC is presented in Algo. 1.

## 3 Experiments

### 3.1 Experimental Settings

**Datasets.** Following [27], we evaluate the node classification performance of the proposed SFGC method on Cora, Citeseer [61], and Ogbn-arxiv [17] under the transductive setting, on Flickr [66] and Reddit [16] under the inductive setting. For all datasets under two settings, we use the public splits and setups for fair comparisons. We consider three condensation ratios ($r$) for each dataset. Concretely, $r$ is the ratio of condensed node numbers $rN(0 < r < 1)$ to large-scale node numbers $N$. In the transductive setting, $N$ represents the number of nodes in the entire large-scale graph, while in the inductive setting, $N$ indicates the number of nodes in the training sub-graph of the whole large-scale graph. The dataset statistic details are shown in Appendix C.

**Baselines & Implementations.** We adopt the following baselines for comprehensive comparisons [27]: graph coarsening method [20], graph coreset methods, *i.e.*, Random, Herding [60], and K-Center [51], the graph-based variant DC-Graph of general dataset condensation method DC [77], which is introduced by [27], graph dataset condensation method GCOND [27] and its variant GCOND-X without utilizing the graph structure. The whole pipeline of our experimental evaluation can be divided into two stages: (1) the condensation stage: synthesizing condensed graph-free data, where we use the classical and commonly-used GCN model [61]; (2) the condensed graph-free data test stage: training a certain GNN model (default with GCN) by the obtained condensed graph-free data from the first stage and testing the GNN on the test set of the large-scale graph with repeated 10 times. We report the average transductive and inductive node classification accuracy (ACC%) with standard deviation (std). Following [27], we use the two-layer GNN with 256 hidden units as the defaulted setting. Besides, we adopt the K-center [51] features to initialize our condensed node attributes for stabilizing the training process. Additional hyper-parameter setting details are listed in Appendix E.

Table 1: Node classification performance (ACC%±std) comparison between condensation methods and other graph size reduction methods with different condensation ratios. (Best results are in bold, and the second-bests are underlined.)

| Datasets | Ratio (r) | Other Graph Size Reduction Baselines | | | | Condensation Methods | | | | Whole Dataset |
|---|---|---|---|---|---|---|---|---|---|---|
| | | Coarsening [20] | Random [60] | Herding [60] | K-Center [51] | DC-Graph [77] | GCOND-X [27] | GCOND [27] | SFGC (ours) | |
| Citeseer | 0.9% | $52.2_{\pm0.4}$ | $54.4_{\pm4.4}$ | $57.1_{\pm1.5}$ | $52.4_{\pm2.8}$ | $66.8_{\pm1.5}$ | $71.4_{\pm0.8}$ | $70.5_{\pm1.2}$ | $\mathbf{71.4}_{\pm0.5}$ | $71.7_{\pm0.1}$ |
| | 1.8% | $59.0_{\pm0.5}$ | $64.2_{\pm1.7}$ | $66.7_{\pm1.0}$ | $64.3_{\pm1.0}$ | $66.9_{\pm0.9}$ | $69.8_{\pm1.1}$ | $\underline{70.6}_{\pm0.9}$ | $\mathbf{72.4}_{\pm0.4}$ | |
| | 3.6% | $65.3_{\pm0.5}$ | $69.1_{\pm0.1}$ | $69.0_{\pm0.1}$ | $69.1_{\pm0.1}$ | $66.3_{\pm1.5}$ | $69.4_{\pm1.4}$ | $\underline{69.8}_{\pm1.4}$ | $\mathbf{70.6}_{\pm0.7}$ | |
| Cora | 1.3% | $31.2_{\pm0.2}$ | $63.6_{\pm3.7}$ | $67.0_{\pm1.3}$ | $64.0_{\pm2.3}$ | $67.3_{\pm1.9}$ | $75.9_{\pm1.2}$ | $\underline{79.8}_{\pm1.3}$ | $\mathbf{80.1}_{\pm0.4}$ | $81.2_{\pm0.2}$ |
| | 2.6% | $65.2_{\pm0.6}$ | $72.8_{\pm1.1}$ | $73.4_{\pm1.0}$ | $73.2_{\pm1.2}$ | $67.6_{\pm3.5}$ | $75.7_{\pm0.9}$ | $\underline{80.1}_{\pm0.6}$ | $\mathbf{81.7}_{\pm0.5}$ | |
| | 5.2% | $70.6_{\pm0.1}$ | $76.8_{\pm0.1}$ | $76.8_{\pm0.1}$ | $76.7_{\pm0.1}$ | $67.7_{\pm2.2}$ | $76.0_{\pm0.9}$ | $\underline{79.3}_{\pm0.3}$ | $\mathbf{81.6}_{\pm0.8}$ | |
| Ogbn-arxiv | 0.05% | $35.4_{\pm0.3}$ | $47.1_{\pm3.9}$ | $52.4_{\pm1.8}$ | $47.2_{\pm3.0}$ | $58.6_{\pm0.4}$ | $\underline{61.3}_{\pm0.5}$ | $59.2_{\pm1.1}$ | $\mathbf{65.5}_{\pm0.7}$ | $71.4_{\pm0.1}$ |
| | 0.25% | $43.5_{\pm0.2}$ | $57.3_{\pm1.1}$ | $58.6_{\pm1.2}$ | $56.8_{\pm0.8}$ | $59.9_{\pm0.3}$ | $\underline{64.2}_{\pm0.4}$ | $63.2_{\pm0.3}$ | $\mathbf{66.1}_{\pm0.4}$ | |
| | 0.5% | $50.4_{\pm0.1}$ | $60.0_{\pm0.9}$ | $60.4_{\pm0.8}$ | $60.3_{\pm0.4}$ | $59.5_{\pm0.3}$ | $63.1_{\pm0.5}$ | $\underline{64.0}_{\pm0.4}$ | $\mathbf{66.8}_{\pm0.4}$ | |
| Flickr | 0.1% | $41.9_{\pm0.2}$ | $41.8_{\pm2.0}$ | $42.5_{\pm1.8}$ | $42.0_{\pm0.7}$ | $46.3_{\pm0.2}$ | $45.9_{\pm0.1}$ | $\underline{46.5}_{\pm0.4}$ | $\mathbf{46.6}_{\pm0.2}$ | $47.2_{\pm0.1}$ |
| | 0.5% | $44.5_{\pm0.1}$ | $44.0_{\pm0.4}$ | $43.9_{\pm0.9}$ | $43.2_{\pm0.1}$ | $45.9_{\pm0.1}$ | $45.0_{\pm0.2}$ | $\mathbf{47.1}_{\pm0.1}$ | $47.0_{\pm0.1}$ | |
| | 1% | $44.6_{\pm0.1}$ | $44.6_{\pm0.2}$ | $44.4_{\pm0.6}$ | $44.1_{\pm0.4}$ | $\underline{45.8}_{\pm0.1}$ | $45.0_{\pm0.1}$ | $47.1_{\pm0.1}$ | $\mathbf{47.1}_{\pm0.1}$ | |
| Reddit | 0.05% | $40.9_{\pm0.5}$ | $46.1_{\pm4.4}$ | $53.1_{\pm2.5}$ | $46.6_{\pm2.3}$ | $88.2_{\pm0.2}$ | $\underline{88.4}_{\pm0.4}$ | $88.0_{\pm1.8}$ | $\mathbf{89.7}_{\pm0.2}$ | $93.9_{\pm0.0}$ |
| | 0.1% | $42.8_{\pm0.8}$ | $58.0_{\pm2.2}$ | $62.7_{\pm1.0}$ | $53.0_{\pm3.3}$ | $89.5_{\pm0.1}$ | $\underline{89.3}_{\pm0.1}$ | $89.6_{\pm0.7}$ | $\mathbf{90.0}_{\pm0.3}$ | |
| | 0.2% | $47.4_{\pm0.9}$ | $66.3_{\pm1.9}$ | $71.0_{\pm1.6}$ | $58.5_{\pm2.1}$ | $\mathbf{90.5}_{\pm1.2}$ | $88.8_{\pm0.4}$ | $\underline{90.1}_{\pm0.5}$ | $89.9_{\pm0.4}$ | |

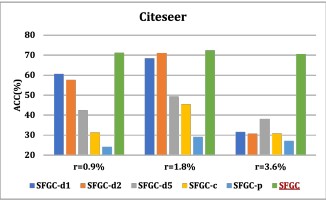
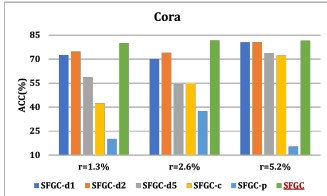
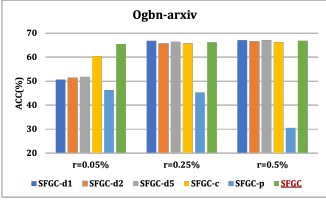

Figure 3: Comparisons between five variants of synthesizing graph structures *vs.* our structure-free SFGC (discrete $k$-nearest neighbor ($k$NN) structure variants: SFGC-d1 ($k = 1$), SFGC-d2 ($k = 2$), and SFGC-d5 ($k = 5$), continuous graph structure variant: SFGC-c, parameterized graph structure variant: SFGC-p).

## 3.2 Experimental Results

**Performance of SFGC on Node Classification.** We compare the node classification performance between SFGC and other graph size reduction methods, especially the graph condensation methods. The overall performance comparison is listed in Table 1. Generally, SFGC achieves the best performance on the node classification task with 13 of 15 cases (five datasets and three condensation ratios for each of them), compared with all other baseline methods, illustrating the high quality and expressiveness of the condensed graph-free data synthesized by our SFGC. More specifically, the better performance of SFGC than GCOND and its structure-free variant GCOND-X experimentally verifies the superiority of the proposed method. We attribute such superiority to the following two aspects regarding the condensation stage. First, the long-term parameter distribution matching of our SFGC works better than the short-term gradient matching in GCOND and GCOND-X. That means capturing the long-range GNN learning behaviors facilitates to holistically imitate GNN's training process, leading to comprehensive knowledge transfer from the large-scale graph to the small-scale condensed graph-free data. Second, the structure-free paradigm of our SFGC enables more compact small-scale graph-free data. For one thing, it liberates the optimization process from triple-level objectives, alleviating the complexity and difficulty of condensation. For another thing, the obtained optimal condensed graph-free data compactly integrates node attribute contexts and topology structure information. Furthermore, on Cora and Citeseer, SFGC synthesizes better condensed graph-free data that even exceeds the whole large-scale graph dataset. These results confirm that SFGC is able to break the information limitation under the large-scale graph and effectively synthesize new, small-scale graph-free data as an optimal representation of the large-scale graph.

**Effectiveness of Structure-free Paradigm in SFGC.** The proposed SFGC introduces the structure-free paradigm without condensing graph structures in graph condensation. To verify the effectiveness of the structure-free paradigm, we compare the proposed SFGC with its variants, which synthesize graph structures in the condensation process. Specifically, we evaluate the following three different methods of synthesizing graph structures with five variants of SFGC: (1) discrete $k$-nearest neighbor ($k$NN) structures calculated by condensed node features under $k = (1, 2, 5)$, corresponding to the variants SFGC-d1, SFGC-d2, and SFGC-d5; (2) cosine similarity based continuous graph structures

Table 2: Performance across different GNN architectures.

| Datasets (Ratio) | Methods | MLP | GAT [56] | APPNP [14] | Cheby [7] | GCN [61] | SAGE [16] | SGC [63] | Avg. |
|---|---|---|---|---|---|---|---|---|---|
| Citeseer ($r = 1.8\%$) | DC-Graph [77] | 66.2 | - | 66.4 | 64.9 | 66.2 | 65.9 | 69.6 | 66.6 |
| | GCOND-X [27] | 69.6 | - | 69.7 | 70.6 | 69.7 | 69.2 | 71.6 | 70.2 |
| | GCOND [27] | 63.9 | 55.4 | 69.6 | 68.3 | 70.5 | 66.2 | 70.3 | 69.0 |
| | SFGC (**ours**) | **71.3** | **72.1** | **70.5** | **71.8** | **71.6** | **71.7** | **71.8** | **71.5** |
| Cora ($r = 2.6\%$) | DC-Graph [77] | 67.2 | - | 67.1 | 67.7 | 67.9 | 66.2 | 72.8 | 68.3 |
| | GCOND-X [27] | 76.0 | - | 77.0 | 74.1 | 75.3 | 76.0 | 76.1 | 75.7 |
| | GCOND [27] | 73.1 | 66.2 | 78.5 | 76.0 | 80.1 | 78.2 | 79.3 | 78.4 |
| | SFGC (**ours**) | **81.1** | **80.8** | **78.8** | **79.0** | **81.1** | **81.9** | **79.1** | **80.3** |
| Ogbn-arxiv ($r = 0.25\%$) | DC-Graph [77] | 59.9 | - | 60.0 | 55.7 | 59.8 | 60.0 | 60.4 | 59.2 |
| | GCOND-X [27] | 64.1 | - | 61.5 | 59.5 | 64.2 | 64.4 | 64.7 | 62.9 |
| | GCOND [27] | 62.2 | 60.0 | 63.4 | 54.9 | 63.2 | 62.6 | 63.7 | 61.6 |
| | SFGC (**ours**) | **65.1** | **65.7** | **63.9** | **60.7** | **65.1** | **64.8** | **64.8** | **64.3** |
| Flickr ($r = 0.5\%$) | DC-Graph [77] | 43.1 | - | 45.7 | 43.8 | 45.9 | 45.8 | 45.6 | 45.4 |
| | GCOND-X [27] | 42.1 | - | 44.6 | 42.3 | 45.0 | 44.7 | 44.4 | 44.2 |
| | GCOND [27] | 44.8 | 40.1 | **45.9** | 42.8 | **47.1** | 46.2 | **46.1** | **45.6** |
| | SFGC (**ours**) | **47.1** | **45.3** | 40.7 | **45.4** | 47.1 | **47.0** | 42.5 | 45.0 |
| Reddit ($r = 0.1\%$) | DC-Graph [77] | 50.3 | - | 81.2 | 77.5 | 89.5 | 89.7 | 90.5 | 85.7 |
| | GCOND-X [27] | 40.1 | - | 78.7 | 74.0 | 89.3 | 89.3 | **91.0** | 84.5 |
| | GCOND [27] | 42.5 | 60.2 | 87.8 | 75.5 | 89.4 | 89.1 | 89.6 | 86.3 |
| | SFGC (**ours**) | **89.5** | **87.1** | **88.3** | **82.8** | **89.7** | **90.3** | 89.5 | **88.2** |

calculated by condensed node features, corresponding to the variant SFGC-c; (3) parameterized graph structure learning module with condensed node features adapted by [27], corresponding to the variant SFGC-p. We conduct experiments on three transductive datasets under nine condensation ratios for each graph structure synthesis variant and the proposed SFGC. The results are presented in Fig. 3. In general, the proposed SFGC achieves the best performance over various graph structure synthesis methods, and these results empirically verify the effectiveness of the proposed structure-free condensation paradigm. More specifically, for discrete $k$-nearest neighbor ($k$NN) structure variants, different datasets adapt different numbers of $k$-nearest neighbors under different condensation ratios, which means predefining the value of $k$ can be very challenging. For example, Citeseer dataset has better performance with $k = 1$ in SFGC-d1 under $r = 0.9\%$ than SFGC-d2 and SFGC-d5, but under $r = 1.8\%$, $k = 2$ in SFGC-d2 performs better than others two. Besides, for continuous graph structure variant SFGC-c, it generally cannot exceed the discrete graph structure variants, except for Ogbn-arxiv dataset under $r = 0.05\%$. And the parameterized variant SFGC-p almost fails to synthesize satisfied condensed graphs under the training trajectory meta-matching scheme. The superior performance of SFGC to all structure-based methods demonstrates the effectiveness of its structure-free paradigm.

**Effectiveness of Graph Neural Feature Score in SFGC.** We compare the learning time between GNN iterative training *vs.* our proposed GNTK-based closed-form solutions of $\gamma_{gnf}$. Note that the iterative training evaluation strategy mandates the complete training of a GNN model from scratch at each meta-matching step, hence, we calculate its time that covers all training epochs under the best test performance for fair comparisons. Typically, for Flickr dataset ($r = 0.1\%$), our proposed $\gamma_{gnf}$ based GNTK closed-form solutions takes only 0.015s for dynamic evaluation, which significantly outperforms the iterative training evaluation with 0.845s. The superior performance can also be observed in Ogbn-arxiv dataset ($r = 0.05\%$) with 0.042s of our $\gamma_{gnf}$, compared with 4.264s of iterative training, illustrating our SFGC's high dynamic evaluation efficiency. More results and analysis of our proposed $\gamma_{gnf}$ in GNTK-based closed-form solutions can be found in Appendix.

**Generalization Ability of SFGC across Different GNNs.** We evaluate and compare the generalization ability of the proposed SFGC and other graph condensation methods. Concretely, we test the node classification performance of our synthesized graph-free data (condensed on GCN) with seven different GNN architectures: MLP, GAT [56], APPNP [14], Cheby [7], GCN [61], SAGE [16], and SGC [63]. It can be generally observed that the proposed SFGC achieves outstanding performance over all tested GNN architectures, reflecting its excellent generalization ability. This is because our method reduces the graph structure to the identity matrix, so that the condensed graph node set can no longer be influenced by different convolution operations of GNNs along graph structures, enabling it consistent and good performance with various GNNs.

More experimental analysis and discussions, including the effects of different ranges of long-term meta-matching, the performance on downstream unsupervised graph clustering task, visualization of our condensed structure-free node set, as well as time complexity analysis, are detailed in Appendix E.

# 4 Conclusion

This work proposes a novel Structure-Free Graph Condensation paradigm, named `SFGC`, to distill the large-scale graph into the small-scale graph-free node set without graph structures. Under the structure-free learning paradigm, the training trajectory meta-matching scheme and the graph neural feature score measured dynamic evaluation work collaboratively to synthesize small-scale graph-free data with superior effectiveness and good generalization ability. Extensive experimental results and analysis under large condensation ratios confirm the superiority of the proposed `SFGC` method in synthesizing excellent small-scale graph-free data. It can be anticipated that our work would bridge the gap between academic GNNs and industrial MLPs by synthesizing small-scale, graph-free data to address graph data scalability, while retaining the expressive performance of graph learning. Our method works on condensing the number of nodes in a single graph at the node level, and we will explore extending it to condense the number of graphs in a graph set at the graph level in the future. We will also explore the potential of unifying graphs and large language models [44] for the graph condensation task.

## Acknowledgment

In this work, S. Pan was supported by an Australian Research Council (ARC) Future Fellowship (FT210100097), and M. Zhang was supported by National Natural Science Foundation of China (NSFC) grant (62306084). This research is partially sponsored by the U.S. National Science Foundation through Grant No IIS-2302786.

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

# Appendix

This is the appendix of our work: **'Structure-free Graph Condensation: From Large-scale Graphs to Condensed Graph-free Data'**. In this appendix, we provide more details of the proposed SFGC in terms of related works, potential application scenarios, dataset statistics, method analysis, and experimental settings with some additional results.

## A    Related Works

**Dataset Distillation (Condensation)** aims to synthesize a small typical dataset that distills the most important knowledge from a given large target dataset, such that the synthesized small dataset could serve as an effective substitution of the large target dataset for various scenarios [30, 49], *e.g.*, model training and inference, architecture search, and continue learning. Typically, DD [59] and DC-KRR [39] adopted the meta-learning framework to solve bi-level distillation objectives through calculating meta-gradients. In contrast, DC [77], DM [76], and MTT [4] designed surrogate functions to avoid unrolled optimization through the gradient matching, feature distribution matching, and training trajectory matching, respectively, where the core idea is to effectively mimic the large target dataset in the synthesized small dataset. Except for the image data condensed by the above-mentioned works, GCOND [27] first extended the online gradient matching scheme in DC [77] to structural graph data, along with parameterized graph structure learning module to synthesize condensed edge connections. Furthermore, DosCond [26] proposed single-step gradient matching to synthesize graph nodes, with a probabilistic graph model to condense structures on the graph classification task. In this work, we eliminate the process of synthesizing graph structures and propose a novel structure-free graph condensation paradigm, to distill the large-scale graph to the small-scale graph-free data, leading to the easier optimization process of condensation. Meanwhile, the structure-free characteristic allows condensed data better generalization ability to different GNN architectures.

**Graph Size Reduction** aims to reduce the graph size to fewer nodes and edges for effective and efficient GNN training, including graph sampling [66, 6], graph coreset [47, 60], graph sparsification [1, 5], graph coarsening [3, 28], and recently rising graph condensation [27, 9, 26]. Concretely, graph sampling methods [66, 6] and graph coreset methods [47, 60] sample or select the subset of nodes and edges from the whole graph, such that the information of the derived sub-graph is constrained by the whole large-scale graph, which considerably limits the expressiveness of the size-reduced graph. Moreover, graph sparsification methods [1, 5] and graph coarsening methods [3, 28] reduce the number of edges and nodes by simplifying the edge connections and grouping node representations of the large-scale graph, respectively. The core idea of both sparsification and coarsening is to preserve specific large-scale graph properties (*e.g.*, spectrum and principle eigenvalues) in the sparse and coarsen small graph. The preserved graph properties in the small-scale graph, however, might not be suitable for downstream GNN tasks. In contrast, our work focuses on graph condensation to directly optimize and synthesize the small-scale condensed data, which breaks information constraints of the large-scale graph and encourages consistent GNN test performance.

## B    Potential Application Scenarios

We would like to highlight the significance of graph condensation task to various application scenarios within the research field of dataset distillation/condensation, while comprehensive overviews can be found in survey works [30, 59]. Specifically, we present several potential scenarios where our proposed structure-free graph condensation method could bring benefits:

**Graph Neural Architecture Search.** Graph neural architecture search (GraphNAS) aims to develop potential and expressive GNN architectures beyond existing human-designed GNNs. By automatically searching in a space containing various candidate GNN architecture components, GraphNAS could derive powerful and creative GNNs with superior performance on specific graph datasets for specific tasks [80, 81, 46, 13, 18]. Hence, GraphNAS needs to repeatedly train different potential GNN architectures on the specific graph dataset, and ultimately selects the optimal one. When in the large-scale graph, this would incur severe computation and memory costs. In this case, searching on our developed small-scale condensed graph-free data, a representative substitution of the large-scale graph, could significantly benefit for saving many computation costs and accelerating new GNN architecture development in GraphNAS research field.

**Privacy Protection.** Considering the outsourcing scenario of graph learning tasks, the original large-scale graph data is not allowed to release due to privacy, for example, patients expect to use GNNs for medical diagnosis without their personal medical profiles being leaked [52, 11]. In this case, as a compact and representative substitution, the synthesized small-scale condensed graph could be used to train GNN models, so that the private information of the original graph data can be protected. Besides, considering the scenario that over-parameterized GNNs might easily memorize training data, inferring the well-trained models could cause potential privacy leakage issue. In this case, we could release a GNN model trained by the synthesized small-scale condensed graph, so that the model avoids explicitly training on the original large-scale graph and consequently helps protect its data privacy.

**Adversarial Robustness.** In practical applications, GNNs might be attacked with disrupted performance, when attackers impose adversarial perturbations to the original graph data [68], for instance, poisoning attacks on graph data [53, 15, 84], where attackers attempt to alter the edges and nodes of training graphs of a target GNN. Training on poisoned graph data could significantly damage GNNs' performance. In this case, given a poisoned original training graph, graph condensation could synthesize a new condensed graph from it, which we use to train the target GNN would achieve comparable test performance with that trained by the original training graph before being poisoned. Hence, the new condensed graph could eliminate adversarial samples in the original poisoned graph data with great adversarial robustness, so that using it to train a GNN would not damage its performance for inferring test graphs.

**Continual learning.** Continual learning (CL) aims to progressively accumulates knowledge over a continuous data stream to support future learning while maintaining previously learned information [45, 12, 65]. One of key challenges of CL is catastrophic forgetting [31, 83], where knowledge extracted and learned from old data/tasks are easily forgotten when new information from new data/tasks are learned. Some works have studied that data distillation/condensation is an effective solution to alleviate catastrophic forgetting [8, 48, 50, 62], where the distilled and condensed data is taken as representative summary stored in a replay buffer that is continually updated to instruct the training of subsequent data/tasks.

To summarize, graph condensation task holds great promise and is expected to bring significant benefits to various graph learning tasks and applications. By producing compact, high-quality, small-scale condensed graph data, graph condensation has the potential to enhance the efficiency and effectiveness of future graph machine learning works.

## C   Dataset Details

We provide the details of the original dataset statistics in Table A1. Moreover, we also compare the statistics of our condensed graph-free data with GCOND [27] condensed graphs in Table A2. It can be observed that both GCOND [27] and our proposed SFGC significantly reduce the numbers of nodes and edges from large-scale graphs, as well as the data storage. Importantly, our proposed SFGC directly reduces the number of edges to 0 by eliminating graphs structures in the condensation process, but with superior node attribute contexts integrating topology structure information.

Table A1: Details of dataset statistics.

| Datasets | #Nodes | #Edges | #Classes | #Features | Train/Val/Test |
|---|---|---|---|---|---|
| Cora | 2,708 | 5,429 | 7 | 1,433 | 140/500/1000 |
| Citeseer | 3,327 | 4,732 | 6 | 3,703 | 120/500/1000 |
| Ogbn-arxiv | 169,343 | 1,166,243 | 40 | 128 | 90,941/29,799/48,603 |
| Flickr | 89,250 | 899,756 | 7 | 500 | 44,625/22312/22313 |
| Reddit | 232,965 | 57,307,946 | 41 | 602 | 15,3932/23,699/55,334 |

## D   More Analysis of Structure-free Paradigm

In this section, we theoretically analyze the rationality of the proposed structure-free paradigm from the views of statistical learning and information flow, respectively.

Table A2: The statistic comparison of condensed graphs by GCOND [27] and condensed graph-free data by our SFGC.

| Dataset | Citeseer ($r = 1.8\%$) | | | Cora ($r = 2.6\%$) | | | Ogbn-arxiv ($r = 0.25\%$) | | | Flickr ($r = 0.5\%$) | | | Reddit ($r = 0.1\%$) | | |
|---|---|---|---|---|---|---|---|---|---|---|---|---|---|---|---|
| Methods | Whole | GCOND [27] | SFGC (ours) | Whole | GCOND [27] | SFGC (ours) | Whole | GCOND [27] | SFGC (ours) | Whole | GCOND [27] | SFGC (ours) | Whole | GCOND [27] | SFGC (ours) |
| Accuracy | 70.7 | 70.5 | 72.4 | 81.5 | 79.8 | 81.7 | 71.4 | 63.2 | 66.1 | 47.1 | 47.1 | 47.0 | 94.1 | 89.4 | 90.0 |
| #Nodes | 3,327 | 60 | 60 | 2,708 | 70 | 70 | 169,343 | 454 | 454 | 44,625 | 223 | 223 | 153,932 | 153 | 153 |
| #Edges | 4,732 | 1,454 | 0 | 5,429 | 2,128 | 0 | 1,166,243 | 3,354 | 0 | 218,140 | 3,788 | 0 | 10,753,238 | 301 | 0 |
| Sparsity | 0.09% | 80.78% | - | 0.15% | 86.86% | - | 0.01% | 3.25% | - | 0.02% | 15.23% | - | 0.09% | 2.57% | - |
| Storage | 47.1MB | 0.9MB | 0.9MB | 14.9MB | 0.4MB | 0.4MB | 100.4MB | 0.3MB | 0.2MB | 86.8MB | 0.5MB | 0.4MB | 435.5MB | 0.4MB | 0.4MB |

**The View of Statistical Learning.** We start from the graph condensation optimization objective of synthesizing graphs structures in Eq. (1) of the main submission. Considering its inner loops $\boldsymbol{\theta}_{\mathcal{T}'} = \arg\min_{\boldsymbol{\theta}} \mathcal{L}\left[\text{GNN}_{\boldsymbol{\theta}}\left(\mathbf{X}', \mathbf{A}'\right), \mathbf{Y}'\right]$ with $\mathbf{A}' = \text{GSL}_{\psi}\left(\mathbf{X}'\right)$, it equals to learn the conditional probability $Q(\mathbf{Y}' \mid \mathcal{T}')$ given the condensed graph $\mathcal{T}' = (\mathbf{X}', \mathbf{A}', \mathbf{Y}')$ as

$$
\begin{aligned}
Q(\mathbf{Y}' \mid \mathcal{T}') &\approx \sum_{\mathbf{A}' \in \psi(\mathbf{X}')} Q(\mathbf{Y}' \mid \mathbf{X}', \mathbf{A}') Q(\mathbf{A}' \mid \mathbf{X}') \\
&= \sum_{\mathbf{A}' \in \psi(\mathbf{X}')} Q(\mathbf{X}', \mathbf{A}', \mathbf{Y}')/Q(\mathbf{X}', \mathbf{A}') \cdot Q(\mathbf{X}', \mathbf{A}')/Q(\mathbf{X}') \\
&= \sum_{\mathbf{A}' \in \psi(\mathbf{X}')} Q(\mathbf{X}', \mathbf{A}', \mathbf{Y}')/Q(\mathbf{X}') \\
&= Q(\mathbf{X}', \mathbf{Y}')/Q(\mathbf{X}') = Q(\mathbf{Y}' \mid \mathbf{X}'),
\end{aligned}
\tag{8}
$$

where we simplify the notation of graph structure learning module $\text{GSL}_{\psi}$ as parameterized $\psi\left(\mathbf{X}'\right)$. As can be observed, when the condensed graph structures are learned from the condensed nodes as $\mathbf{A}' \in \psi\left(\mathbf{X}'\right)$, the optimization objective of the conditional probability is not changed, while its goal is still to solve the posterior probability $Q(\mathbf{Y}' \mid \mathbf{X}')$. In this way, eliminating graph structures to conduct structure-free condensation is rational from the view of statistical learning. By directly synthesizing the graph-free data, the proposed SFGC could ease the optimization process and directly transfer all the informative knowledge of the large-scale graph to the condensed graph node set without structures. Hence, the proposed SFGC conducts more compact condensation to derive the small-scale graph-free data via Eq. (6) of the main manuscript, whose node attributes already integrate implicit topology structure information.

**The View of Information Flow.** For training on large-scale graphs to obtain offline parameter trajectories, we solve the node classification task on $\mathcal{T} = (\mathbf{X}, \mathbf{A}, \mathbf{Y})$ with a certain GNN model as

$$
\boldsymbol{\theta}_{\mathcal{T}}^* = \arg\min_{\boldsymbol{\theta}} \mathcal{L}_{\text{cls}}\left[\text{GNN}_{\boldsymbol{\theta}}(\mathbf{X}, \mathbf{A}), \mathbf{Y}\right],
\tag{9}
$$

where $*$ denotes the optimal training parameters that build the training trajectory distribution $P_{\boldsymbol{\Theta}_{\mathcal{T}}}$. The whole graph information, *i.e.*, node attributes $\mathbf{X}$ and topology structures $\mathbf{A}$ are both embedded in the latent space of GNN network parameters. Hence, the large-scale graph information flows to GNN parameters as $(\mathbf{X}, \mathbf{A}) \Rightarrow P_{\boldsymbol{\Theta}_{\mathcal{T}}}$. In this way, by meta-sampling in the trajectory distribution, Eq. (4) and Eq. (5) in the main manuscript explicitly transfer learning behaviors of the large-scale graph to the parameter space $\widetilde{\boldsymbol{\theta}}_{\mathcal{S}}$ of $\text{GNN}_{\mathcal{S}}$ as $P_{\boldsymbol{\Theta}_{\mathcal{T}}} \Rightarrow \widetilde{\boldsymbol{\theta}}_{\mathcal{S}}$. As a result, the informative knowledge of the large-scale graphs, *i.e.*, node attributes and topology structure information $(\mathbf{X}, \mathbf{A})$, would be comprehensively transferred as $(\mathbf{X}, \mathbf{A}) \Rightarrow P_{\boldsymbol{\Theta}_{\mathcal{T}}} \Rightarrow \widetilde{\boldsymbol{\theta}}_{\mathcal{S}}$. In this way, we could identify the critical goal of graph condensation is to further transfer the knowledge in $\widetilde{\boldsymbol{\theta}}_{\mathcal{S}}$ to the output condensed graph data as:

$$
\begin{aligned}
(\mathbf{X}, \mathbf{A}) &\Rightarrow P_{\boldsymbol{\Theta}_{\mathcal{T}}} \Rightarrow \boldsymbol{\Theta}_{\mathcal{S}} \Rightarrow \mathcal{T}' = (\mathbf{X}', \mathbf{A}'), \quad \text{GC.} \\
(\mathbf{X}, \mathbf{A}) &\Rightarrow P_{\boldsymbol{\Theta}_{\mathcal{T}}} \Rightarrow \boldsymbol{\Theta}_{\mathcal{S}} \Rightarrow \mathcal{S} = (\widetilde{\mathbf{X}}), \quad \text{SFGC.}
\end{aligned}
\tag{10}
$$

where GC and SFGC are corresponding to the existing graph condensation and the proposed structure-free graph condensation, respectively.

Hence, from the view of information flow, we could observe that condensing structures would not inherit more information from the large-scale graph. Compared with GC which formulates the condensed graph into nodes and structures, the proposed SFGC directly distills all the large-scale graph knowledge into the small-scale graph node set without structures. Consequently, the proposed SFGC conducts more compact condensation to derive the small-scale graph-free data, which implicitly encodes the topology structure information into the discriminative node attributes.

Table A3: Running time comparison (seconds) of the proposed `SFGC` and GCOND [27] for 50 epochs with a single GeForce RTX 3080 GPU.

| Ogbn-arxiv | r=0.05% | r=0.25% | r=0.5% |
|---|---|---|---|
| GCOND[27] | 296.34 | 442.58 | 885.58 |
| SFGC (**ours**) | 101.07 | 183.54 | 150.35 |

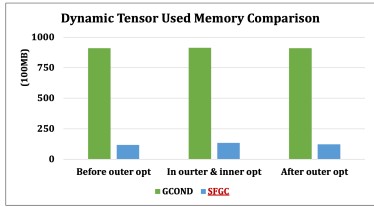

Figure A1: Comparison of the dynamic tensor used memory cost between online short-range gradient matching method GCOND [27] and our proposed `SFGC`.

# E    More Experimental Settings and Results

## E.1    Time Complexity Analysis & Dynamic Memory Cost Comparison

We first analyze the time complexity of the proposed method and compare the running time between our proposed `SFGC` and GCOND [27].

Let the number of GCN layers be $L$, the large-scale graph node number be $N$, the small-scale condensed graph node number be $N'$, the the feature dimension be $d$, the time complexity of calculating training trajectory meta-matching objective function is about $TK\mathcal{O}(LN'd^2 + LN'd)$, for each process of the forward, backward, and training trajectory meta-matching loss calculation, where $T$ denotes the number of iterations and $K$ denotes the meta-matching steps. Note that the offline expert training stage costs an extra $TK\mathcal{O}(LEd + LNd^2)$ on the large-scale graph, where $E$ is the number of edges.

In contrast, for GCOND, it has at least $TK\mathcal{O}(LN'^2d + LN'd) + TK\mathcal{O}(N'^2d^2)$, and also additional $TK\mathcal{O}(LEd + LNd^2)$ on the large-scale graph, where $K$ denotes the number of different initialization here. It can be observed that our proposed `SFGC` has a smaller time complexity compared to GCOND, which can be mainly attributed to our structure-free paradigm when the adjacency matrix related calculation in $\mathcal{O}(LN'^2d)$ can be avoided. The corresponding comparison of running time in the graph condensation process can be found in Table A3. As can be observed, both results on time complexity and running time could verify the superiority of the proposed `SFGC`.

Moreover, we present the comparison result of the dynamic tensor used memory cost between the online short-range gradient matching method GCOND [27] and our offline long-range meta-matching `SFGC`. As shown in Fig. A1, we consider three stages of optimizing the objective function, *i.e.*, before outer optimization, in the outer and inner optimization, and after outer optimization. It can be observed that the proposed `SFGC` could significantly alleviate heavy online memory and computation costs. This can be attributed to its offline parameter matching schema.

## E.2    Effectiveness of Graph Neural Feature Score in SFGC

To verify the effectiveness of graph neural feature score $\gamma_{\text{gnf}}$ in the proposed `SFGC`, we consider the following two aspects in dynamic evaluation: (1) node classification performance at different meta-matching steps in Table A4; (2) learning time comparison between iterative GNN training and our closed-form $\gamma_{\text{gnf}}$ in Fig. A2.

As shown in Table A4, we select certain meta-matching step intervals, *i.e.*, 1000, 2000, and 3000, for testing their condensed data's performance, which is a commonly-used evaluation strategy for existing methods. Here, we set long-enough meta-matching steps empirically to ensure sufficient learning to expert training trajectory-built parameter distribution. And we compare these interval-step results with the performance of our condensed graph-free data, which is selected at certain steps of

Table A4: Performance of the condensed graph-free data between different meta-matching steps and $\gamma_{\mathrm{gnf}}$ dynamic evaluation selected steps in the proposed SFGC.

| Datasets (Ratio) | Meta-matching Steps | | | $\gamma_{\mathrm{gnf}}$ | |
|---|---|---|---|---|---|
| | 1000 | 2000 | 3000 | ACC | Selected Steps |
| Citeseer ($r = 1.8\%$) | 61.8±3.1 | 64.2±5.2 | - | 72.4±0.4 | 46 |
| Cora ($r = 2.6\%$) | 81.2±0.5 | 81.8±0.7 | - | 81.7±0.5 | 929 |
| Ogbn-arxiv ($r = 0.25\%$) | 64.5±0.8 | 65.8±0.3 | - | 66.1±0.4 | 90 |
| Flickr ($r = 0.5\%$) | 46.3±0.2 | 44.7±0.3 | - | 47.0±0.1 | 200 |
| Reddit ($r = 0.1\%$) | 86.9±0.5 | 89.8±0.3 | 89.9±0.5 | 90.0±0.3 | 2299 |

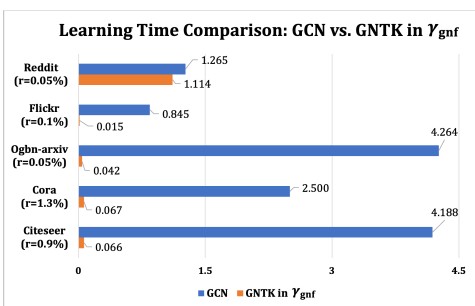

Figure A2: Learning time comparison (seconds) in dynamic evaluation between GNN iterative training and closed-form GNTK in $\gamma_{\mathrm{gnf}}$ of the proposed SFGC.

the meta-matching process according to the metric $\gamma_{\mathrm{gnf}}$. Overall, $\gamma_{\mathrm{gnf}}$ could select optimal condensed graph-free data with superior effectiveness at best meta-matching steps.

For the learning time comparison between GNN iterative training *vs.* GNTK-based closed-form solutions of $\gamma_{\mathrm{gnf}}$ in Fig. A2, we consider the time of GNN iterative training that covers all training epochs under the best test performance for fair comparisons. This is due to the fact that the iterative training evaluation strategy mandates the complete training of a GNN model from scratch at each meta-matching step. For instance, in Flickr dataset ($r = 0.1\%$), we calculate 200 epochs running time, *i.e.*, 0.845s, which is the optimal parameter setting for training GNN under 0.1% condensation ratio. As can be generally observed, for all datasets, the proposed GNTK-based closed-form solutions of $\gamma_{\mathrm{gnf}}$ significantly save the learning time for evaluating the condensed graph-free data in meta-matching, illustrating our SFGC's high dynamic evaluation efficiency.

### E.3 Analysis of Different Meta-matching Ranges

To explore the effects of different ranges of long-term meta-matching, we present the different step combinations of $q$ steps (student) in $\mathrm{GNN}_{\mathcal{S}}$ and $p$ steps (expert) of $\mathrm{GNN}_{\mathcal{T}}$ in Eq. (5) of the main manuscript on Ogbn-arxiv dataset under $r = 0.05\%$. The results are shown in Fig. A3. As can be observed, there exists the optimal step combination of $q$ student steps (600) and expert $p$ steps (1800). Under such a setting, the condensed small-scale graph-free data has the best node classification performance. Moreover, the quality and expressiveness of the condensed graph-free data moderately vary with different step combinations, but the variance is not too drastic.

More detailed settings of hyper-parameters of $q$ steps (student) in $\mathrm{GNN}_{\mathcal{S}}$ and $p$ steps (expert) of $\mathrm{GNN}_{\mathcal{T}}$ in the long-term meta-matching, as well as the meta-matching learning rate (LR) in the outer-level optimization and $\mathrm{GNN}_{\mathcal{S}}$ learning rate (step size) $\zeta$ (Algorithm 1 of the main manuscript) in the inner-level optimization, are listed in Table A5.

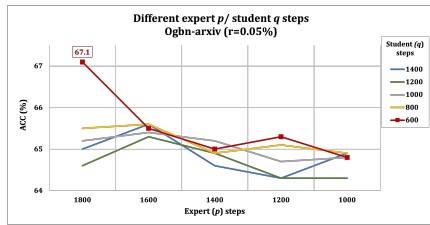

Figure A3: Performance with different step combinations of $q$ student steps and expert $p$ steps on Ogbn-arxiv ($r = 0.05\%$).

Table A5: Hyper-parameters of $p$ expert steps and $q$ student steps with meta-matching learning rate (LR) in the outer-level optimization and $\mathrm{GNN}_{\mathcal{S}}$ learning rate (step size) $\zeta$ in the inner-level optimization.

| Datasets | Ratios ($r$) | $p$ steps (expert) | $q$ steps (student) | Meta-matching LR | $\zeta$ for $\mathrm{GNN}_{\mathcal{S}}$ |
|---|---|---|---|---|---|
| Citeseer | 0.9% | 500 | 200 | 0.0005 | 1.0 |
| | 1.8% | 500 | 200 | 0.001 | 1.0 |
| | 3.6% | 400 | 300 | 0.001 | 1.0 |
| Cora | 1.3% | 1500 | 400 | 0.0001 | 0.5 |
| | 2.6% | 1200 | 500 | 0.0001 | 0.5 |
| | 5.2% | 2000 | 500 | 0.0001 | 0.5 |
| Ogbn-arxiv | 0.05% | 1800 | 600 | 0.2 | 0.2 |
| | 0.25% | 1900 | 1200 | 0.1 | 0.1 |
| | 0.5% | 1900 | 1000 | 0.1 | 0.1 |
| Flickr | 0.1% | 700 | 600 | 0.1 | 0.3 |
| | 0.5% | 900 | 600 | 0.01 | 0.2 |
| | 1% | 900 | 900 | 0.02 | 0.2 |
| Reddit | 0.05% | 900 | 900 | 0.02 | 0.5 |
| | 0.1% | 900 | 900 | 0.05 | 0.5 |
| | 0.2% | 900 | 900 | 0.2 | 0.2 |

### E.4 Performance on Graph Node Clustering Task

Taking graph node clustering as the downstream task, we verified that, our condensed graph-free data, synthesized based on the node classification task, can be effectively utilized for other graph machine learning tasks, demonstrating the applicability of our condensed data. The experimental results are shown in Table A6 and Table A7 below.

Concretely, we use our condensed graph-free data, which is generated using GNN classification experts, to train a GCN model. Then, the trained GCN model conducts clustering on the original large-scale graph. The clustering results in percentage on Cora and Citeseer datasets are shown by four commonly-used metrics, including clustering accuracy (C-ACC), Normalized Mutual Information (NMI), F1-score (F1), and Adjusted Rand Index (ARI).

Table A6: Performance comparison on Cora in terms of graph node clustering. Best results are in bold and the second best are with underlines.

| Clusterings on Cora | C-ACC | NMI | F1 | ARI |
|---|---|---|---|---|
| K-means | 50.0 | 31.7 | 37.6 | 23.9 |
| VGAE [29] | 59.2 | 40.8 | 45.6 | 34.7 |
| ARGA [42] | 64.0 | 44.9 | 61.9 | 35.2 |
| MGAE [58] | 68.1 | 48.9 | 53.1 | **56.5** |
| AGC [75] | 68.9 | 53.7 | 65.6 | 44.8 |
| DAEGC [57] | 70.4 | 52.8 | 68.2 | 49.6 |
| SUBLIME [35] | 71.3 | **54.2** | 63.5 | 50.3 |
| SFGC (**ours**) ($r = 1.3\%$) | 70.5 | 51.9 | 71.0 | 43.7 |
| SFGC (**ours**) ($r = 2.6\%$) | 69.4 | 51.3 | 70.1 | 42.2 |
| SFGC (**ours**) ($r = 5.2\%$) | **71.8** | 53.0 | **73.1** | 43.8 |

Table A7: Performance comparison on Citeseer in terms of graph node clustering. Best results are in bold and the second best are with underlines.

| Clusterings on Citeseer | C-ACC | NMI | F1 | ARI |
|---|---|---|---|---|
| K-means | 54.4 | 31.2 | 41.3 | 28.5 |
| VGAE [29] | 39.2 | 16.3 | 27.8 | 10.1 |
| ARGA [42] | 57.3 | 35.0 | 54.6 | 34.1 |
| MGAE [58] | 66.9 | 41.6 | 52.6 | 42.5 |
| AGC [75] | 67.0 | 41.1 | 62.5 | 41.5 |
| DAEGC [57] | 67.2 | 39.7 | 63.6 | 41.0 |
| SUBLIME [35] | **68.5** | **44.1** | 63.2 | **43.9** |
| SFGC (**ours**) ($r = 0.9\%$) | 64.9 | 38.1 | 63.6 | 37.3 |
| SFGC (**ours**) ($r = 1.8\%$) | 66.5 | 39.4 | **64.9** | 39.7 |
| SFGC (**ours**) ($r = 3.6\%$) | 65.3 | 37.6 | 63.4 | 38.0 |

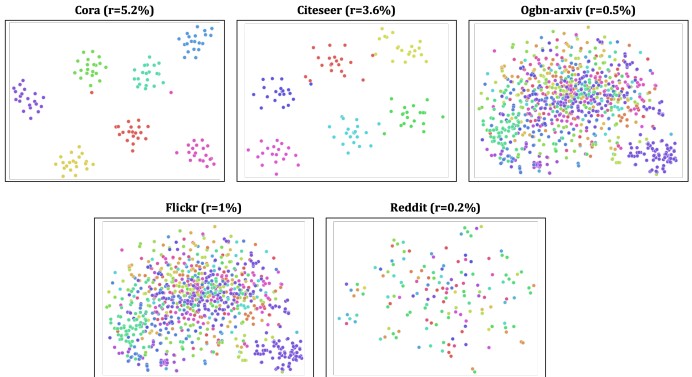

Figure A4: Visualization of t-SNE on condensed graph-free data by SFGC.

As can be observed, our condensed graph enables the GNN model to achieve comparable results with many graph node clustering baseline methods, even though we do not customize the optimization objective targeting node clustering task in the condensation process. These results could justify that: (1) the condensed graph-free data that is synthesized based on GNN classification experts, could also work well in other tasks, even without task-specific customization in the condensation process; (2) the condensed graph-free data contains adequate information about the original large-scale graph, which can be taken as the representative and informative substitution of the original large-scale graph, reflecting the good performance of our proposed method in graph condensation.

### E.5 Visualization of Our Condensed Graph-free Data

we present t-SNE [55] plots of the condensed graph-free data of our proposed SFGC under the minimum condensation ratios over all datasets in Fig. A4. The condensed graph-free data shows a well-clustered pattern over Cora and Citeseer. In contrast, on larger-scale datasets with larger condensation ratios, we can also observe some implicit clusters within the same class. These results show that the small-scale graph-free data synthesized by our method has discriminative and representative node attributes that capture comprehensive information from large-scale graphs.

