# OpenReview forum: "Structure-free Graph Condensation: From Large-scale Graphs to Condensed Graph-free Data"
_NeurIPS.cc/2023/Conference — NeurIPS 2023 spotlight_

### Official Review · Reviewer_4KcB · 2023-06-28

**Soundness:** 4 excellent
**Presentation:** 3 good
**Contribution:** 4 excellent
**Rating:** 7
**Confidence:** 4

**Summary:**

This work proposes a structure-free graph condensation paradigm to distill a large-scale graph into a small-scale graph node set without explicit graph structures. The node attributes of the obtained small-scale condensed graph-free data could encode topology structure information. And the condensed node set could serve as a substitution to replace the large-scale graph for training GNNs and achieves comparable performance on the test set.
The method contains two techniques, (1) the parameter matching schema under imitation learning, and (2) a dynamic evaluation schema with a GNTK score. The experimental results could verify its claims and show performance effectiveness.


**Strengths:**

S1-[originality]: This work's most remarkable part is its structure-free condensation paradigm, which removes the graph structure by encoding it into node attributes, so that the obtained condensed graph-free data only contains a set of informative nodes. I think this paper would inspire many interesting questions for future researches and topics, when only use a set of nodes to represent the whole large-scale graph for training but obtain comparable test results.

S2-[clarity]: Overall, the question descriptions, contributions, techniques, and experimental results of this paper are described clearly and soundly.

first, the concept of structure-free graph condensation and its relevant scenarios have been clearly defined and exemplified.

second, the contributions and techniques including the training trajectory matching with online GNN parameters and the GNTK-based dynamic score are clear and sound.
This paper could ease three-level optimization to a bi-level optimization without condensed graph structures, which is a novel and interesting structure-free graph condensation pattern.
The GNTK-based dynamic evaluation score enables the closed-form solution of GNN evaluation and sounds novel to me.

third, the experimental results compared with the whole graph dataset training results could support this paper's arguments. An ablation study verifies the effectiveness of the structure-free paradigm. There is another interesting part that the results across different GNN architectures show surprisingly good generalization ability in terms of the condensed node set. This is an inspirable result and deserves future explorations.

S3-[significance]: this work on graph condensation is an interesting problem to reduce the effects brought by large-graph data scale and quantity, and it might be helpful for reducing calculations in real-world applications.


**Weaknesses:**

There are only some technical details that are not well presented here.

W1: how to choose the condensation ratios like Cora with 0.9% and Citeseer with 1.3%? More details and explanations should be involved.

W2: how to deal with the condensed graph labels Y? Is it generated according to the original large-scale graph's classes and each class examples?

W3: are the student steps and teacher steps empirical hyperparameters or learnable parameters that need to be optimized in the condensation process? More details should be given. Besides, this work provides some interesting results, and more discussions of experimental findings should be given, for instance, why the synthesized graph-free data has good generalization ability.

**Questions:**


Please provide more details corresponding to the weakness.

**Limitations:**

Yes, the authors have mentioned limitations, no negative social impact.

---

> ### Author Rebuttal · Authors · 2023-08-10
>
> **Response to Reviewer 4KcB**
>
> Thank you for taking the time to review my work and providing your valuable feedback. We are so encouraged by your positive comments on our originality and clear clarity and we appreciate your recognition of our research significance. The following are our detailed responses. We are expecting these could be helpful in answering your questions.
>
> **W1: Details of condensation ratio**:
> The condensation ratio is chosen based on the labeling rates of the benchmark graph datasets for training, that is the proportion of labeled training nodes in relation to all nodes. For example, in Cora and Citeseer, their training set labeling rates are 5.2% and 3.6% (for 20 labeled samples per class), respectively, and we choose the set according to labeling percentage as {25%, 50%, 100%}, corresponding to {1.3%, 2.6%, 5.2%} for Cora, and {0.9%, 1.8%, 3.6%} for Citeseer. Hence, the 5.2% and 3.6% would be the maximum ratios for Cora and Citeseer, respectively.
>
> **W2: Condensed graph labels Y**:
> As mentioned in Lines 133-134 of the main submission, for all graph condensation tasks, 'label $Y^{\prime}$ of the small-scale condensed graph are pre-defined based on the class distribution of the label space $Y$ in the large-scale graph.'
> For instance, given Flickr dataset, which has $N = 44,625$ nodes from $C=7$ classes (labeled from '0' to '6') in its training graph.
> Among $N = 44,625$ nodes, we have $N_{C1}=4,321$ nodes labeled '1', $N_{C2}=3,164$ nodes labeled '2', and $N_{C0}+N_{C1}+\cdots+N_{C6}=N$. In this case, for the $r=0.1$% condensation ratio, the synthesized condensed graph-free data would have $N_{C1}\times r = 4,321\times 0.1$% $\approx 4$ nodes labeled '1' and $N_{C2}\times r=3,164\times 0.1$% $\approx 3$ nodes labeled '2', and so on for the other classes.
>
> **W3: Student/ teacher step setting and experimental finding discussions.**
> Student steps and teacher steps are empirical hyperparameters with no need to be optimized. We have provided their settings in Table A5 of the Appendix.
> We appreciate your recognition in terms of the good generalization ability of synthesized graph-free data condensed by our proposed SFGC. One of our contributions is to overcome the poor generalization ability issue in existing graph condensation methods with our proposed structure-free paradigm. The main reason that leads to such good generalization ability can be that different GNN architectures mainly differ in convolution operations along graph structures, our proposed structure-free paradigm would minimize the impact of different convolution operations on graph structures by only learning with a condensed graph node set, leading to consistent and reliable performance across various GNN architectures.  Thanks for your suggestion and we will add more discussions in our revised version.

---

> > ### Comment · Reviewer_4KcB · 2023-08-17
> >
> > Thank you for the response and my confusion has been clarified. I keep my score.

---

> > > ### Comment · Area_Chair_eDKq · 2023-08-18
> > >
> > > Dear Reviewers,
> > >
> > > I hope this message finds you well. First and foremost, thank you all for your valuable contributions to the NeurIPS review process.
> > >
> > > As we approach a critical phase of the review cycle, we've noticed that there are still pending responses to authors' rebuttals and specific replies from some reviewers. These responses are pivotal in ensuring a comprehensive and constructive review process for all submissions.
> > >
> > > In light of this, we kindly request those of you who have outstanding responses to kindly prioritize completing them as soon as possible. Your prompt attention to this matter would be greatly appreciated, as it will help us maintain the momentum of the review process and ensure that authors receive timely feedback.
> > >
> > > If you encounter any challenges or require additional assistance, please don't hesitate to reach out to us. We are here to support you and facilitate a smooth and efficient review process.
> > >
> > > Once again, we express our gratitude for your dedication to NeurIPS and your role as a reviewer. Your efforts play a vital role in shaping the quality and impact of the conference.
> > >
> > > Thank you for your attention, and we look forward to your continued collaboration.
> > >
> > > Thanks,
> > > AC

---

### Official Review · Reviewer_PLzT · 2023-07-06

**Soundness:** 2 fair
**Presentation:** 2 fair
**Contribution:** 2 fair
**Rating:** 5
**Confidence:** 2

**Summary:**

This paper studies an interesting problem. The authors proposes a new paradigm for reducing the size of large-scale graphs without explicit graph structures. The proposed SFGC, encodes topology structure information into node attributes in synthesized graph-free data. Extensive experiments demonstrate the effectiveness of the proposed method compred with existing graph condensation methods.

**Strengths:**

- The paper proposes a novel SFGC approach.
- This paper studies the effectiveness and generalization ability issues in graph condensation.
- This paper provides theoretical illustrations of the proposed structure-free graph condensation paradigm from the views of statistical learning and information flow, respectively.

**Weaknesses:**

- The work fails to provide a clear motivation for why it is important or required to reduce the size of big graphs lacking explicit graph structures.
- The paper could benefit from a more thorough discussion of the limitations and potential future directions of the proposed approach.
- Without having access to the source code, it is challenging to reproduce the results. The experimental setting is described, however it would be beneficial to have access to the source code to guarantee that the findings can be reproducible.

If the main concern about the reproducibility is solved, I am willing to increase the score.


**Questions:**

- How computationally efficient is the SFGC strategy compared to the current graph condensation techniques?
- How effective and robust is the SFGC method on real-world graphs with noisy or insufficient data?

**Limitations:**

The authors could benefit from a more thorough discussion of the limitations and potential negative societal impacts of their work, as well as potential ways to mitigate these issues.

---

> ### Author Rebuttal · Authors · 2023-08-10
>
> **Response to Reviewer PLzT**
>
> We sincerely appreciate the time and effort you dedicated to reviewing our work. We have carefully considered your comments and suggestions. Following the instructions for the rebuttal, we have sent the source code via an anonymous link to the AC. We appreciate it if you obtain the link from AC. The source code will be made public upon acceptance. The following are our detailed responses. We are expecting these could be helpful in answering your questions.
>
> **W1: Motivation for why it is “important or required” to reduce the size of big graphs lacking explicit graph structures**:
> In the general context of dataset condensation and graph size reduction, our proposed SFGC shares a similar motivation that reducing the size of big graphs into small counterparts could help in reducing storage costs and accelerating GNN model development progress, as mentioned in Lines 25-30 of the main submission.
> Moreover, the motivations for our proposed ‘Structure-free’ graph condensation pattern (without explicit graph structures) are in the following aspect: (1) simplifying existing complex ‘triple-level’ optimization (‘with structure’) into effective ‘bi-level’ progress (‘structure-free’) to improve graph condensation effectiveness (Ref. Lines 60-66 of the main submission); (2) alleviating the limitation of condensed graph data on certain graph convolution operations that depends on graph structures to improve synthesized graph data generalization ability.(Ref. Lines 67-70 of the main submission).
> Hence, even without explicit graph structures, our proposed SFGC encodes topology structure information into the node attributes in the synthesized graph-free data (Ref. Lines 76-78 of the main submission), achieving comparable test performance with the original large-scale graph.
> Besides, the importance and necessity of our proposed structure-free graph condensation can also be reflected in various potential applications in practice (Ref. Section-B of Appendix), covering: graph neural architecture search, privacy protection, adversarial robustness, and continue learning.
>
> **W2: More thorough discussion of the limitations and potential future direction.**
> As mentioned at the end of our main submission (ref. Lines 358-360),  our proposed SFGC mainly works on node-level condensation by reducing the number of nodes in a single graph, as a result, our condensed graph-free data has limited ability on graph-level tasks, which requires multiple graphs to supervise GNN training.
> In light of this, for potential future direction, we would like to further explore more on the graph-level condensation problem, which should simultaneously reduce “the number of nodes“ and “the number of graphs“ in the large-scale graph collection. The proposed long-term training trajectory meta-matching scheme would be considered for multiple graph condensation scenarios, and the core challenge would be how to jointly incorporate the node-level and graph-level complexity and diversity into the condensation progress for high-quality condensed data.
> Besides, condensing graph data for various downstream graph learning tasks is another promising research direction, for instance, condensing large-scale graph data to a small-scale counterpart for improving GNN online serving performance.
>
> **Q1: How computationally efficient is the SFGC strategy compared to the current graph condensation techniques?**:
> As mentioned in our Appendix, in Table A3, Figure A1, and Sec.E.1,  we have provided the (1) Running time comparison, (2) Dynamic tensor used memory cost, and (3) Theoretical Time Complexity Analysis, respectively, to illustrate the computation efficiency of our proposed SFGC and existing graph condensation method GCOND [14].
> In summary, our proposed method has (1) 5x less running time (SFGC: 150.35s vs. GCOND [14]: 885.58s) on Ogbn-arxiv condensation; (2) significantly low dynamic memory usage (SFGC: 118.116 vs. GCOND [14] 910.4)*100Mb in the overall optimization progress; and (3) at least less O(LN'^2d) time complexity.
> More detailed results and analysis can be found in our Appendix. These results could reflect the good computational efficiency of our proposed SFGC compared with the current graph condensation method.
>
> **Q2: How effective and robust is the SFGC method on real-world graphs with noisy or insufficient data?**
> For **noisy data**,  we would like to emphasize that, in the context of graph condensation, our proposed SFGC is a “data-centric” method that synthesizes small-scale graph-free data from the original input graph, with the constraint of ensuring comparable test performance on GNNs.
> Hence, if the original graph is noisy and taken as the input for training a certain GCN* model, in the process of graph condensation,  the condensed small-scale graph-free data might learn the distilled comprehensive information (including noisy) from the original graph by imitating the GCN*’s learning behavior. In this case, the noisy information could also be filtered by the graph condensation process.
>
> For **insufficient data**, as the graph condensation technique, the central focus of our proposed SFGC is to distill large-scale graph data into small-scale synthetic graph-free data as its training substitution. Hence, when given graph data is ‘insufficient’ (not in large-scale quantity), there might be no necessity to further condense it in practical scneario.

---

> > ### Comment · Reviewer_PLzT · 2023-08-18
> >
> > Thanks for your response.
> > The response has addressed my concerns. I would like to raise my score.

---

> ### Author Response · Authors · 2023-08-17
> **Gentle Reminder to Reviewer PLzT**
>
> Dear Reviewer PLzT,
>
> Thank you again for taking the time to provide valuable feedback on our paper, and we genuinely hope that our responses have adequately addressed your concerns and questions. We would like to kindly inquire whether you have received the anonymous code link from the AC?
>
> We are sincerely looking forward to your response, and we are always open to engaging in further discussions to address any questions or concerns you may have regarding our work.
>
> Thank you and best regards.

---

### Official Review · Reviewer_yKD6 · 2023-07-07

**Soundness:** 4 excellent
**Presentation:** 3 good
**Contribution:** 4 excellent
**Rating:** 6
**Confidence:** 1

**Summary:**

This paper presents a new method to condense a training graph into a smaller number of disconnected nodes, such that a GNN trained on these nodes performs similar to one trained on the original graph at test time.

**Strengths:**

**Originality.** Structure-free condensation has been reported before (GCOND-X), but the proposed method to do so is novel.

**Quality.** The overall quality of the work is good, including the presented techniques, results, tables and plots.

**Clarity.** The paper is mostly clear.

**Significance.** Graph condensation can be very useful under the right application scenarios. I don't see any special significance of aiming for structure-free condensation, except that the method ends up giving better results (e.g. due to easier optimization, as discussed in the paper).

**Weaknesses:**

The paper is not particularly easy to follow.

GCOND-X is not discussed explicitly in the related works, even though it is a structure-free graph condensation method.

Hyper-parameter details have not been provided.

**Questions:**

What does the condensed structure-free data look like in relation to the original training data? Some intuitive visualization would be nice to have.

What are the GNN depths used? It seems that the fidelity of structure-free condensation should go down as more neighbor aggregation steps are involved at test time, as they were not encountered when training on the condensed data.

**Limitations:**

A discussion of limitations is missing. An obvious limitation is that structure-free condensation won't work under ego- and neighbor-embedding separation [1] which is an effective recommendation for heterophilic datasets.

[1] Zhu et. al. "Beyond Homophily in Graph Neural Networks: Current Limitations and Effective Designs." NeurIPS 2020

---

> ### Author Rebuttal · Authors · 2023-08-10
>
> **Response to Reviewer yKD6**
>
> We sincerely appreciate your thoughtful review of our paper. We are glad to hear that you recognize the significance of graph condensation research, as well as encouraging comments for our work. We have carefully considered your comments and suggestions, and the following are our detailed responses. We are expecting these could be helpful in answering your questions.
>
> **W1: Discussion of GCOND’s graphless variant GCOND-X**:
> Our SFGC has different motivations and technical implementations with GCOND-X [14]. GCOND aims to simultaneously learn node attributes and graph structure, 'graphless' variant GCOND-X is not their main goal, but a by-product for ablation study. In contrast, our SFGC directly encodes structure information into more compact condensed node attributes, with comprehensive theoretical analysis and thorough empirical studies in Appendix Sec. D and Sec.3. Besides, our SFGC conducts offline long-term training trajectory meta-matching schema for condensation, where GCOND-X conducts online gradient matching schema. Thanks for your suggestion and we will add these discussions in the final version.
>
> **W2: Hyper-parameter details**: We have provided detailed hyper-parameter settings in Table A5 containing student and teacher steps, meta-matching learning rate, GNN training step size for all 5 datasets with 15 condensation cases.
>
> **Q1: Visualization of the condensed structure-free data in relation to the original training data**:
> Thanks for your suggestion, and the visualization of (a) original Cora dataset and (b) the condensed Cora dataset (r=5.2%) by our proposed SFGC in **Fig. Re1 of the response PDF file** for illustrating their relationship, and we will add this to the final version.
> It can be observed that our proposed method significantly distills the original graph with complex structures (black dense edges ) to reduced small-scale node set without explicit graph structures. Importantly, they share the same class-label space and similar test performance, as illustrated by the experimental results in Table 1 of our main submission.
>
>
> **Q2: What are the GNN depths used? As neighbors (structures) were not encountered when training on the condensed data, whether performance of structure-free condensation would drop at the test time**:
> We use two-layer GCN for condensation progress.
> Even the topology neighbors “were not encountered when training on the condensed data” explicitly, the performance of our proposed SFGC on the test set of large-scale graph would not drop at the test time (as illustrated by the results in Table 1 of our main submission). That is because, our proposed structure-free condensation could enforce the condensed node features to encode topology structure information of the original large-scale graph, and its GNN learning behavior imitation strategy could comprehensively distill the large-scale graph information (both nodes and structures) into small-scale condensed node set.
>
> **Discussed Limitation and potential future direction**: We mentioned the limitation at the end of our main submission (ref. Lines 358-360),  that is our proposed SFGC mainly works on node-level condensation by reducing the number of nodes in a single graph, which has limited ability on graph-level tasks that require multiple graphs to supervise GNN training. In light of this, for potential future direction, we would like to further explore more on the graph-level condensation problem, which should simultaneously reduce “the number of nodes“ and “the number of graphs“ in the large-scale graph collection.
>
> **L1: Heterophilic datasets**: Thank you for suggesting the interesting heterophily graph type. On the heterophilic graph dataset, a possible and straightforward solution under our proposed SFGC can be: we use the heterophilic-GNN, e.g, H2GNN mentioned in [1], as the condensation model (rather than vanilla GCN in our submission) to distill the information of the original heterophilic graph to the condensed graph-free data.  Once the heterophily characteristic of the original heterophilic graph is condensed into the synthesized node attributes, we still use the condensed data to train a GNN model and infer on the test set of the heterophilic graph. However, effectively learning the intricate and diverse heterophily characteristics presents a severe challenge that deserves more future exploration. We deeply value your perceptive insights and we will add the discussion of this interesting research question to the final version.

---

> > ### Comment · Reviewer_yKD6 · 2023-08-19
> >
> > Thank you for the rebuttal. I would like to keep my score.

---

### Official Review · Reviewer_KcU6 · 2023-07-07

**Soundness:** 3 good
**Presentation:** 4 excellent
**Contribution:** 3 good
**Rating:** 6
**Confidence:** 4

**Summary:**

This paper studies the problem of reducing the size of a large graph dataset while preserving task-relevant information.
It introduces a new methodology to distill large-scale real-world graphs into smaller synthetic graph node sets by disregarding graph structures to create condensed graph-free data. The approach involves two key components: a training trajectory meta-matching scheme for effectively synthesizing small-scale graph-free data, and a graph neural feature score metric for evaluating the quality of condensed graph-free data dynamically. Extensive experiments have demonstrated the efficiency and effectiveness of the proposed method.



**Strengths:**

1. The paper addresses an interesting problem in the field of graph condensation, which has significant practical implications.
2. The proposed SFGC methodology exhibits commendable performance and generalization across various graph neural network (GNN) architectures.
3. The utilization of the Graph Neural Tangent Kernel (GNTK) to avoid iterative training of GNNs adds an interesting aspect to the paper.



**Weaknesses:**


Suggestions for Improvement:
1. To further strengthen the paper, it is recommended to demonstrate the benefits of SFGC in practical applications such as neural architecture search, privacy protection, adversarial robustness, or continual learning. Including at least one of these applications would greatly enhance the paper's significance.
2. It would be of interest to clarify how SFGC can benefit neural architecture search for GNNs since it does not generate graph structures, which are essential for GNNs, and different GNNs may require distinct operations over the graph structure.
3. While Figure A2 provides a comparison of the running time between GCN and GNTK, it would be valuable to include a detailed complexity analysis of both methods concerning the number of nodes.
 * Specifically, elaborate on the quadratic complexity of GNTK due to the pairwise kernel matrix calculations and the matrix inversion operation.
  * I guess that is also why on Reddit (r=0.05%) GCN and GNTK exhibit similar running times. What if we further increase r to 0.1%?
4. It would be beneficial to include an empirical comparison with DosCond, as it also aims to accelerate the graph condensation process, to provide a comprehensive evaluation of SFGC.


**Questions:**

1. Complexity comparison of GCN and GNTK
2. What is the exact formulation for $\mathcal{K}$ in Eq. 7? Specifically, what are the values for $\beta$ and $K$.

**Limitations:**

Yes

---

> ### Author Rebuttal · Authors · 2023-08-09
>
> **Response to Reviewer KcU6**
>
> Thanks for sharing your thoughts and questions with us. We greatly appreciate your valuable suggestions on discussing more on the practical application and complexity of our proposed SFGC method. We have taken your suggestions into careful consideration and we have provided detailed responses to your questions below. We hope that these answers will help to address your concerns clearly.
>
> **W1: Demonstrate the benefits of SFGC in practical applications**:
> Thanks for your suggestion. We have given some detailed illustrations about how the benefits of SFGC in Appendix Sec.B Potential Application Scenarios.
> For instance, for graph neural architecture search (GraphNAS), which needs to repeatedly train different potential GNN architectures, our SFGC generated small-scale condensed graph-free data can be taken as a representative substitution of the large-scale graph, significantly benefiting for saving many computation costs and accelerating new GNN development. A more detailed explanation can be seen in responses to W2.
>
> **W2: How SFGC can benefit GNNs neural architecture search**:
> According to the below survey work *[IJCAI-Survey-2021]*, the design of GraphNAS search space can extensively involve: (a) Micro search space with “aggregation functions”, “aggregation weights”, number of attention heads, combining functions, feature dimensionality, Non-linear activation function; (b) Macro search space with layer-wise combination function; (c) the Pooling function and (d) Hyper-parameters.
> Hence, for fine-grained search space, our proposed structure-free can only slightly affect the design of “aggregation functions and weights” in the (a) Micro search space. Considering all different types of aggregation weights in different GNNs are calculated based on the node features, SFGC condensed graph-free data has implicitly encoded topology structure information into the node features. Hence, SFGC could still benefit GraphNAS by designing node-attribute weighted aggregation functions in the search space, as well as all other aspects in (b)-(d) to design new GNN models driven by specific tasks. Thanks for your valuable suggestions and we will add these discussions into the final version.
>
> *[IJCAI-Survey-2021] Ziwei Zhang, Xin Wang, Wenwu Zhu Automated Machine Learning on Graphs: A Survey.*
>
> **W3-Detailed complexity analysis of GCN and GNTK**:
> Given $T$ is the number of GCN training iterations, $N$ and $N^{'}$ are the number of large-scale graph nodes and condensed graph nodes, respectively, $L$ denotes the number of layers, and $F$ denotes feature dimension.
> In our SFGC, we use $L=2$, hence for GCN, we have its complexity dominated by $\mathcal{O}(4TN^{'}F^{2})$. For node-level GNTK calculation, its complexity can be dominated by $\mathcal{O}(4N^{2}N^{'2})$.
> Hence, under certain iterative times $T$ setting, the GCN’s and GNTK’s might have comparable complexity ($TF^{2}$ vs. $N^{'}N^{2}$), which might be a reason for Reddit (r=0.05%) GCN and GNTK have similar running times (GNTK still better).
> When the condensed graph has more nodes, for instance, Reddit further increases r to 0.1%, GCN needs to involve more iterative times $T$, and GNTK also needs to calculate a bigger Kronecker product matrix. It might be hard to make a straightforward comparison, since it is hard to make sure how to set hyperparameters (for instance, $T$, learning rate) to iteratively train GCN on the mid-product condensed graph in the optimization process. And this challenge is our main motivation to leverage close-form GNTK in dynamic evaluation, avoiding iterative GCN training tied to hyperparameter settings. We sincerely thank you for your suggestion, and we will add these discussions in the final version.
>
> **W4-Empirical comparison with DosCond**:
> The node classification performance comparison between our proposed SFGC and DosCond can be seen in Table.Re-KcU6-1, where the DosCond results are from its work. As can be observed, our proposed method still outperforms DosCond.
> DosCond improves the graph condensation process by simplifying short-range gradient matching to the one-step pattern to accelerate condensation, in contrast, our proposed SFGC improves the graph condensation process by simplifying the optimization objective with a structure-free paradigm to obtain high-quality condensed graph-free data. DosCond and our proposed SFGC do have different targets. Thanks for your valuable suggestion, we will add this discussion to our final version.
>
> **Table.Re-KcU6-1. Performance comparison between DosCond and our proposed SFGC.**
>
> | Methods | Cora (r=2.6%) | Citeseer (r=1.8%) | Flickr (r=0.1%) |
> | :--- | :---: | :---: | :---: |
> | DosCond | 80.0 | 71.0 | 46.1 |
> | SFGC (**Ours**) |**81.7** | **72.4** | **46.6** |
>
> **Q2-Formulation of Eq.(7)**: The K in Eq.(7) is the GNTK kernel and its detailed calculation has been illustrated in Eq.(2) of our main submission. The $\beta$ denotes the number of fully-connected layers in calculating GNTK in Eq.(2) and we set it to 2 in our work.

---

> > ### Comment · Reviewer_KcU6 · 2023-08-15
> > **Thanks for the rebuttal**
> >
> > Thank you for the detailed response and some of my concerns have been addressed. I tend to accept this paper and I think my current score is reasonable.

---

### Official Review · Reviewer_8VHW · 2023-07-25

**Soundness:** 3 good
**Presentation:** 3 good
**Contribution:** 2 fair
**Rating:** 5
**Confidence:** 3

**Summary:**

This paper introduces a structure-free graph condensation method designed to distill large-scale graphs into small-scale graph-free data while preserving comparable expressiveness. The proposed method, named SFGC, achieves this by condensing the graph topology into an identity matrix, effectively embedding the structure information into the node attributes. To effectively imitate the GNN training process, SFGC employs a training trajectory meta-matching scheme. Additionally, a graph neural feature scoring technique is used, which dynamically evaluates the quality and relevance of the synthetic graph-free data.

**Strengths:**

- Graph condensation is a crucial research area with many real-world applications. The authors propose a new “structure-free” graph condensation method.
- They provide convincing experimental results and comprehensive discussions overall.
- The idea of using GNTK-based graph neural feature score metric is interesting and effective.
- The paper is clearly written and easy to follow. Supplementary materials also offer valuable additional information regarding the model and its performance behaviors.


**Weaknesses:**

- The authors claim that this is the first work that distills large-scale graphs to small-scale synthetic graph-free data, but the previous work GCOND-X also appears to perform a similar task of distilling large graphs to small-scale graph-free data. Further clarification on this would be needed.
- It appears that the experiments do not precisely determine the extent to which different aspects of the model contribute to performance improvement.
- The effectiveness of the “structure-free paradigm” in SFCG should also be convincingly demonstrated. In Section 3.2, a graph structure is generated from condensed node features and compared with SFCG. However, since the graph structure is already included in the node features, inputting the graph structure and condensed node features into the GNN model again could cause over-smoothing. Therefore, it is unclear whether the performance improvement over existing models is due to the structure-free paradigm or appears to be due to over-smoothing.


**Questions:**

- Could you provide more details on how the three condensation ratios for each dataset used in the experiments were determined?
- I am curious why the performances of the five variants of SFGC in Figure 3 (Citeseer), used for synthesizing graph structures, drop significantly at a condensation ratio (r) of 3.6%, which is even much worse than the case of smaller condensation ratios.
- I would like to know if the number of expert trajectories affects the performance and how the number of expert trajectories were set.
- Claiming that the long-term parameter distribution matching method is superior because the SFGC method outperforms GCOND and GCOND-X in Table 1 might be somewhat hasty. While the superior performance could be due to the consideration of long-range, it seems that finding the optimal condensed graph-free data, as seen in Table A4, also plays a significant role.



**Limitations:**

The authors do not include a discussion on the potential limitations of this study or offer insights into possible future research directions.

- An important consideration is that if SFGC necessitates the pre-training of a GNN on a large-scale graph to obtain a pre-trained training trajectory, it implies that real-world large-scale data must be initially trained. While this process is distinct from the graph condensation pipeline, it nonetheless introduces challenges associated with intensive computational demands. Addressing and suggesting solutions for this could provide valuable directions for future studies.

- I find the potential of employing different GNN models as the condensation network intriguing. As the method of incorporating graph structural information into node features heavily relies on the characteristics of the condensation model, using a different GNN model could potentially alter the properties of the condensed graph and impact the final performance. Therefore, additional experiments with various condensation networks and test networks would offer valuable insights.

---

> ### Author Rebuttal · Authors · 2023-08-10
>
> **Response to Reviewer 8VHW**
>
> We are glad that you recognize the significance of graph condensation. We have carefully considered your thoughtful comments and suggestions, and the following are our detailed responses.
>
> **W1: Clarification on GCOND’s graphless variant GCOND-X**:
> Our SFGC has different motivations and technical implementations with GCOND-X [14]. GCOND aims to simultaneously learn node attributes and graph structure, 'graphless' variant GCOND-X is not their main goal, but a by-product for ablation study. In contrast, our SFGC directly encodes structure information into more compact condensed node attributes, with comprehensive theoretical analysis and thorough empirical studies in Appendix Sec. D and Sec.3. Besides, our SFGC conducts offline long-term training trajectory meta-matching schema for condensation, whereas GCOND-X conducts online gradient matching schema. We will add these discussions in the final version.
>
> **W2: To what extent each aspect of the model contribute to performance improvement**:
> We have three core components in our framework: (C1) training trajectory meta-matching scheme, (C2) graph neural feature score metric, (C3) structure-free paradigm.We have individually analyzed the effectiveness of each one, i.e., Lines 327-336 (C1), Table A4 (C2), and Fig. 3  (C3) in the submission. To summarize the component contributions, we showed **Table.Re 1 in the response PDF file**  for the ablation study in Cora dataset with 2.6% condensation rate.
> As observed,  IDX-1 vs. IDX-3, IDX-3 vs. IDX-4, and IDX-2 vs. IDX-3, verify the effectiveness of C1, C2, and C3, with 5.5%, 0.5, 7.3% improvement, respectively, illustrating the effectiveness of each aspect of our proposed SFGC.
>
> **W3: Whether the effectiveness of SFGC due to the structure-free paradigm or oversmoothing.**
> We have compared our “structure-free paradigm” (w/o structure) vs. other 5 variants (w/ structure) in Sec.3.2 to verify its effectiveness, where the 5 w/ structure variants follow existing graph structure learning methods, i.e., GCOND[14], to synthesize graph structures from condensed node features. Such graph structure learning strategy might have potential over-smoothing when “graph structure is already included in the node features”.  Importantly, this is the main drawback of existing methods and motivates us to propose “structure-free condensation paradigm”, and performance improvement in Fig.3 over other variants (w/ structure) could illustrate the effectiveness of “structure-free paradigm” in SFGC.
>
> **Q1: Details of condensation ratio**: Due to response word limitations, please refer to our response to Reviewer-4KcB, W1 for more details.
>
> **Q2:Synthesizing graph structures drop at a large condensation rate**:
> This observation accurately reflects the limitation of existing graph structure learning based condensation methods, which require optimizing a triple-level condensation objective. In fact, when condensation rate is relatively large 3.6% in Citeseer,  the number of nodes is larger, and the graph structure learning space increases exponentially, making the problem harder to be optimized. Hence, it is intuitive that synthesizing graph structures would drop significantly at a large condensation rate with more nodes. And this also motivates us to propose the “structure-free paradigm’’.
>
> **Q3:The number of expert trajectories $K$**:
> In our submission, $K$ is empirically set as 200. Here, we also conduct a hyperparameter analysis on $K$ in **Table.Re 2 of the response PDF file**. Intuitively, more experts might lead to more guidance on condensation, and fewer experts might limit the model behavior imitations. However, when the number of experts keeps raising from 200 to 300, the performance drops moderately. One potential reason is: the distribution of more expert GNN parameters would be more complex, and accurately computing their expectation to guide condensation would be more difficult. We will add these discussions in the final version.
>
> **Q4: Superiority of long-term parameter distribution matching**
> First, compared to “online short-range gradient matching” in GCOND and GCOND-X, the superiority of our “offline long-term parameter distribution matching” is (1) good condensation performance and (2) saved memory usage (SFGC: 118.116 vs. GCOND [14] 910.4)*100Mb (ref. Figure A1 in the appendix).
> We list the comparison results w/ and w/o the GNTK-based dynamic evaluation strategy in **Table.Re 3 of the response PDF file**. It shows, our SFGC (w/o dynamic evaluation) achieves better performance than GCOND-X and GCOND, verifying the effectiveness of the proposed “offline long-term parameter distribution matching”.
>
> **Discussed Limitation and potential future direction**: We mentioned the limitation at the end of our main submission (ref. Lines 358-360), that is our SFGC has limited ability on graph-level tasks which build on multiple graphs. And we would like to further explore more on the graph-level condensation problem, which should simultaneously reduce the number of nodes and the number of graphs in the large-scale graph collection.
>
> **L1. Suggestion for intensive computation on training real-world large-scale graph data**: We would suggest employing GraphSage [10], GraphSAINT [35], or  ClusterGCN [KDD-19-Cluster-GCN] as the backbone to train super-large graphs for alleviating for intensive computation.
>
> **L2. Employing different GNN condensation networks**: Our work can be straightforwardly extended to employ different GNN models as condensation networks by imitating their training behaviors. Thanks for sharing the thoughtful suggestion, and we will add these limitation discussions to the final version.

---

> > ### Comment · Reviewer_8VHW · 2023-08-17
> >
> > Thank you for the detailed response. Some of my concerns have been addressed and I'd keep my score.

---

### Official Review · Reviewer_U4b1 · 2023-07-26

**Soundness:** 4 excellent
**Presentation:** 3 good
**Contribution:** 3 good
**Rating:** 6
**Confidence:** 2

**Summary:**

This paper proposes a graph dataset condensing algorithm with a main idea of creating a new format of graph representation that does not explicitly include edge information. The authors suggest a new graph kernel and training algorithm with trajectory for online gradient to achieve graph condensation. Experiments demonstrate that the suggested algorithm (named SFGC) outperforms other baselines that condense the graph with explicit edge information in terms of node classification. The authors also showcase the performance of SFGC in terms of generalization ability and empirical learning time efficiency.


**Strengths:**

One of the main strengths of this paper is introducing an interesting method for compressing graph datasets. The idea of removing edge information explicitly seems strong, but it appears plausible and well-founded, akin to non-negative matrix factorization with constraints of non-negativity. The simplicity of the idea allows other researchers to easily adapt it to their own graph-related research. Furthermore, the authors present numerous experimental results demonstrating that the suggested framework outperforms the baseline models.


**Weaknesses:**

I have some questions, so I hope to listen to the answers from the authors.

It is hard to see Figure 3.

**Questions:**

- Can SFGC be used for edge classification tasks, such as edge prediction?
- I am curious why using SFGC is better than using the whole dataset in Cora data (Table 1). In lines 303 to 304, the authors mentioned this phenomenon, but it would be beneficial to see the reason backed with examples.


**Limitations:**

It is challenging to find the limitation section. Could you please indicate the part where the limitations of the proposed method are discussed?

---

> ### Author Rebuttal · Authors · 2023-08-06
>
> **Response to Reviewer U4b1**
>
> We sincerely appreciate the time and effort the reviewer dedicated to reviewing our work, and we are pleased to learn that our proposed SFGC is interesting and well-founded to the reviewer. The following are our detailed responses to the reviewer’s thoughtful comments and suggestions. And we will refine our manuscript to make Fig.3 more clear.
>
> **Q1: Can SFGC be used for edge classification tasks, such as edge prediction?**:
> Yes, our SFGC can be also used for edge classification tasks, e.g.,  edge prediction, by using edge prediction loss (BCE loss) as the optimization objection. More specifically, according to the definition of graph condensation mentioned in Lines 45-52 of our main submission, the proposed SFGC method condenses large-scale graph data into small-scale condensed graph-free data, so that the small-scale condensed data could achieve comparable ‘test performance’ as the large-scale graph when training the same GNN model. Thus, the proposed method will remain effective for different tasks regarding the ‘test performance’.
>
> **Q2: More analysis of Cora dataset results**:
> The reason SFGC condensed graph-free data has a performance that even exceeds the whole large-scale graph dataset on Cora and Citeseer is mainly attributed to the following aspects:
>
> SFGC is a data-centric method in a generative way. The parameterized node features are continuously updated and optimized in the whole condensation progress of imitating GNN’s learning behavior, which means we have very extensive space to seek optimal condensed graph-free data, resulting in models outperforming the one trained from the original graph. Besides, the long-term training trajectory meta-matching technique allows us to receive comprehensive knowledge as informative supervision from extensive GNN expert training processes by learning their parameter distribution, which further contributes the superior performance of our condensed graph from the original graph. Thanks for your suggestion and we will add such discussions in the final version.
>
> **Discussed Limitation and potential future direction**: We mentioned the limitation at the end of our main submission (ref. Lines 358-360),  that is our proposed SFGC mainly works on node-level condensation by reducing the number of nodes in a single graph, which has limited ability on graph-level tasks that require multiple graphs to supervise GNN training. In light of this, for potential future direction, we would like to further explore more on the graph-level condensation problem, which should simultaneously reduce “the number of nodes“ and “the number of graphs“ in the large-scale graph collection.

---

### Author Rebuttal · Authors · 2023-08-09

**Common response to all reviewers**:

We thank all reviewers for their thorough review and valuable suggestions. We are delighted that our contributions have been positively acknowledged, including:

**(1) Novel and interesting problem of structure-free graph condensation paradigm for practical application scenarios (@All Reviewers!)**

**(2) Effective long-term training trajectory meta-matching framework with the bi-level optimization (Reviewer 4KcB,Reviewer yKD6)**

**(3) Interesting and effective GNTK-based evaluation metric with graph neural feature score (Reviewer 8VHW, Reviewer 4KcB, Reviewer KcU6)**

**(4) Numerous and convincing experimental results with superior performance effectiveness. (@All Reviewers!)**

**(5) Good generalization ability of our synthesized graph over cross architectures (Reviewer KcU6, Reviewer PLzT, Reviewer 4KcB)**

We greatly appreciate all the positive comments and valuable suggestions for our work. These comments encourage us to continue our efforts in advancing this new promising graph condensation research area for real-world applications. More detailed responses are as follows. We hope our responses address all weaknesses and questions! Please let us know if there is any concern. We have considered your thoughtful suggestions, and have modified accordingly to improve the manuscript in the final version.

---

### Decision · Program_Chairs · 2023-09-21

**Decision:**

Accept (spotlight)

**Comment:**

Reviewers unanimously offer affirmative feedback on the manuscript. They concur that the research addresses a novel problem. Both the proposed methodology and the chosen evaluation metrics demonstrate effectiveness, while the experimental results are convincing.